

**Molecular distribution and stable carbon isotopic compositions of**
**dicarboxylic acids and related SOA from biogenic sources in the**
**summertime atmosphere of Mt. Tai in the North China Plain**
Jingjing Meng[1,3], Gehui Wang[2,3,4*], Zhanfang Hou[1,3], Xiaodi Liu[1], Benjie Wei[1], Can Wu[3],
Cong Cao[3], Jiayuan Wang[3], Jianjun Li[3], Junji Cao[3], Erxun Zhang[1], Jie Dong[1], Jiazhen Liu[1]
[1] School of Environment and Planning, Liaocheng University, Liaocheng 252000, China
[2] Key Laboratory of Geographic Information Science of the Ministry of Education, School
of Geographic Sciences, East China Normal University, Shanghai 200062, China
[3] State Key Laboratory of Loess and Quaternary Geology, Key Lab of Aerosol Chemistry
and Physics, Institute of Earth Environment, Chinese Academy of Sciences, Xi'an 710075,
China
[4] School of Human Settlements and Civil Engineering, Xi'an Jiaotong University, Xi'an
710049, China
*Corresponding author: Prof. Gehui Wang
E-mail address: wanggh@ieecas.cn, or ghwang@geo.ecnu.edu.cn





**Abstract**: Molecular distributions and stable carbon isotopic ($\delta^{13}$C values) compositions of
dicarboxylic acids and related SOA in $PM_{2.5}$ aerosols collected on a day/night basis at the
summit of Mt. Tai (1534 m a.s.l) in the summer of 2016 were analyzed to investigate the
sources and photochemical aging process of organic aerosols in the forested highland region
of North China Plain. The molecular distributions of dicarboxylic acids and related SOA are
characterized by the dominance of oxalic acid ($C_2$), followed by malonic ($C_3$), succinic ($C_4$)
and azelaic ($C_9$) acids. The concentration ratios of $C_2/C_4$, diacid-C/OC and $C_2$/total diacids
are larger in daytime than in nighttime, suggesting that the daytime aerosols are more
photochemically aged than those in nighttime due to the higher temperatures and stronger
solar radiation. Both ratios of $C_2/C_4$ ($R^2 > 0.5$) and $C_3/C_4$ ($R^2 > 0.5$) correlated strongly with the
ambient temperature, indicating that SOA in the mountaintop atmosphere are mainly derived
from the photochemical oxidation of local emissions rather than long-range transport. The
mass ratios of $C_9/C_6$, $C_9$/Ph, Gly/mGly and the strong linear correlation of major
dicarboxylic acids and related SOA with biogenic precursors further suggest that aerosols in
this region are mainly originated from biogenic sources (i.e., tree emissions).

$C_2$ concentrations correlated well with aerosol pH, indicating that particle acidity favors

the organic acid formation. The stable carbon isotopic compositions ($\delta^{13}$C) of the
dicarboxylic acids are higher in daytime than in nighttime with the highest value
($-16.5 \pm 1.9$‰) found for $C_2$ and the lowest value ($-25.2 \pm 2.7$‰) found for $C_9$. An increase in
$\delta^{13}$C values of $C_2$ along with increases in $C_2$/Gly and $C_2$/mGly ratios was observed, largely
due to the isotopic fractionation during photochemical degradation of the precursors.

**Keywords**: Dicarboxylic acids; Glyoxal and methylglyoxal; Secondary organic aerosols
(SOA); Biogenic sources; Formation mechanisms






## 1. Introduction

Secondary organic aerosols (SOA) accounts for a substantial fraction (20-90%) of the total $PM_{2.5}$ mass in the troposphere, and up to 80% of which are water-soluble (Hallquist et al., 2009; Kroll and Seinfeld, 2008). Due to the low vapor pressures and high hygroscopicity (approximately less than $10^{-7}$ Pa) (Bilde et al., 2015; Ehn et al., 2014), dicarboxylic acids and related compounds are ubiquitously found in atmospheric waters and particles (Kawamura and Bikkina, 2016; Sorooshian et al., 2007a). Because of the water-soluble and hygroscopic properties, dicarboxylic acids and related compounds play important roles in atmospheric aqueous chemistry and influence radiative forcing of aerosols via acting as cloud condensation nuclei (CCN) (Hoque et al., 2017; Wang et al., 2016; Zhang et al., 2016).

Although they can be emitted directly from sources such as incomplete combustion of fossil fuels (Kawamura and Kaplan, 1987) and biomass burning (Kawamura et al., 2013a, b; Narukawa et al., 1999), atmospheric dicarboxylic acids and related compounds are largely produced by photochemical oxidation of unsaturated fatty acids, PAHs (Kawamura et al., 1996) , cyclic alkanes and other compounds (Kawamura and Usukura, 1993). Oxalic acid ($C_2$) is the smallest and the most abundant dicarboxylic acid (Wang et al., 2009, 2015). Modeling studies and cloud measurements have suggested that $C_2$ are largely produced from aqueous-phase oxidation of less oxygenated organic precursors such as glyoxal (Gly), methyglyoxal (mGly) and pyruvic acid (Pyr) in clouds or wet aerosols as well as the photochemical breakdown of longer-chain dicarboxylic acids (Wang et al., 2012, 2015).

There is a growing consensus on highlighting the significance of oxalic acid and related SOA formation from the photochemical oxidation of anthropogenic/biogenic volatile organic compounds (VOCs) via the aqueous phase in clouds or liquid water content (LWC)-enriched aerosols from many field observations and laboratory experiments as well as modeling studies (Bikkina et al., 2017; Cheng et al., 2017; Ervens et al., 2014; Fu et al., 2008; Lim et al., 2005; Miyazaki et al., 2009; Mochizuki et al., 2017). A ubiquitous layer of dicarboxylic acids was found above the clouds by aircraft measurements in US, indicating



that organic acids are important CCN in the free troposphere (Sorooshian et al., 2007a, b).
Compared to the aerosols in lowland areas, alpine aerosols have a more important influence
on cloud formation, because they are more accessible to clouds due to higher elevation. Mt.
Tai is an independent peak located in the center of the North China Plain (NCP), one of the
severest air- polluted regions in the world (Wang et al., 2009; Yang et al., 2017). A few
studies have been performed on the molecular distributions and sources of dicarboxylic
acids at Mt. Tai, but most of them were conducted during the wheat straw burning period
(Kawamura et al., 2013a, b; Zhu et al., 2018), and very few information on dicarboxylic
acids in Mt. Tai during July and August is available when biogenic activity of vegetation is
dominant. Therefore, it is necessary to investigate the chemistry of SOA during the typical
summer season in this highland area.
The stable carbon isotopic composition ($\delta^{13}C$) of specific organic acids can provide
very useful information on the sources and photochemical aging of organic aerosols,
because the isotopic fractionation of carbon occurs upon chemical reactions or phase transfer
(Pavuluri and Kawamura, 2016; Zhang et al., 2016). To our best knowledge, the stable
carbon isotopic compositions of dicarboxylic acids and related SOA in mountainous regions
have not been reported before. In the current work, we first investigated the diurnal
variations in molecular distributions and stable carbon isotopic compositions of dicarboxylic
acids and related compounds. Then we discussed the impact of temperature (T), relative
humidity (RH), particle acidity ($pH_{IS}$), liquid water content (LWC) and $O_3$ concentration on
oxalic acid and related SOA to explore their sources and formation mechanism in the
forested highland of the North China Plain.
**2. Experimental section**
**2.1 Aerosol sampling**
PM$_{2.5}$ samples were collected at the Meteorological Observation Station of Mt. Tai,
which is located at the summit of Mt. Tai (36.25° N, 117.10°E; 1534 m a.s.l.) in the North
China Plain (Fig. 1). About 80% of the mountainous land is covered by vegetation known to
comprise 989 species, which is densely wooded in summer (Fu et al., 2010). PM$_{2.5}$ samples



were collected from July 20 to August 20, 2016 each lasting for 12h on a day/night basis
using a mid-volume air sampler (KC-120H, Qingdao Laoshan Company, China) equipped
with prebaked (450°C, 8 h) quartz fiber filters (Whatman, USA) at an airflow rate of 100 L
min$^{-1}$. The daytime samples were collected from 8:00 to 20:00, while nighttime samples
were collected from 20:00 to 8:00 of the next day. Field blank samples were also collected
by mounting the blank filter onto the sampler for 15 min without sucking any air before and
after the campaign. A total of 57 samples (daytime: 28; nighttime: 29) were collected during
the campaign. After sampling, each filter was sealed in an aluminum foil bag and stored at
−20°C prior to laboratory analyses. Moreover, the concentration of ozone was
simultaneously monitored at the side by an UV absorption analyzer (Model 49C, Thermo
Electron Corporation).
**2.2 Chemical analyses**
**2.2.1 Dicarboxylic acids, ketocarboxylic acids, and α-dicarbonyls**
Dicarboxylic acids, ketocarboxylic acids and α-dicarbonyls in PM$_{2.5}$ were determined
using the method described by previous studies (Kawamura et al.,1996; Meng et al., 2013,
2014). Briefly, one half of the filter was cut into pieces and extracted with pure Milli-Q
water under ultrasonication for three times each for 15 min. The water extracts were
concentrated to near dryness and then reacted with 14% BF$_3$/$n$-butanol at 100°C for 1 hr to
form butyl esters/dibutoxy acetals. After derivatization, $n$-hexane was added and washed
with pure water for three times. Finally, the hexane layer was concentrated to 200 μL and
determined using a capillary gas chromatography (GC; HP 6890) coupled with a
split/splitless injector and a flame ionization detector (FID). The GC oven temperature was
programmed from 50 (2 min) to 120°C at 15°C  min$^{-1}$, and then to 300 at 5°C min$^{-1}$ with a
final isothermal hold at 300°C for 16 min. Peak identification was performed by comparing
the GC retention time with that of authentic standards and confirmed by mass spectrum of
the sample using a GC-mass spectrometer (GC-MS). Recoveries of the target compounds
were 80% for oxalic acid and 85% to 110% for other species. The target compounds in the



field blank samples were lower than 4% of those in the ambient samples. Data presented
here were corrected for both field blanks and recoveries.
**2.2.2 Stable carbon isotope composition of dicarboxylic acids and related SOA**
The stable carbon isotopic compositions ($\delta^{13}C$) of shorter-chain dicarboxylic acids and
related SOA were measured using the method developed by Kawamura and Watanabe
(2004). Briefly, the $\delta^{13}C$ values of the derivatized samples above were determined by gas
chromatography–isotope ratio mass spectrometry (GC-IR-MS; Thermo Fisher, Delta V
Advantage). The $\delta^{13}C$ values were then calculated for free organic acids using an isotopic
mass balance equation based on the measured $\delta^{13}C$ values of derivatives and the derivatizing
agent ($BF_3$/n-butanol) (Kawamura and Watanabe, 2004). To ensure the analytical error of
the $\delta^{13}C$ values less than 0.2 ‰, each sample was measured three times. The $\delta^{13}C$ data
reported here are averaged values of the triplicate measurements.
**2.2.3 Elemental carbon (EC), organic carbon (OC), inorganic ions, water-soluble**
**organic carbon (WSOC), aerosol liquid water content (LWC), and particle in-situ pH**
**($pH_{IS}$).**
Briefly, EC and OC in the $PM_{2.5}$ samples were determined by using DRI Model 2001
Carbon Analyzer following the Interagency Monitoring of Protected Visual Environments
(IMPROVE) thermal/optical reflectance (TOR) protocol (Chow et al., 2004). As for the
measurement of inorganic ions and WSOC, an aliquot of the sample filters was extracted
with 30 mL Milli-Q water using an ultrasonic bath for three times each for 15min, and
filtered through PTFE filters to remove the particles and filter debris. The water extract was
then separated into two parts. One part was analyzed for inorganic ions using an ion
chromatography (Dionex 600, Dionex, USA), and the other part of the water extract was
used to determine WSOC using a Total Carbon Analyzer (TOC-L CPH, Shimadzu, Japan).
As for the calculation of aerosol liquid water content (LWC) and particle in-situ pH ($pH_{IS}$),
the Aerosol Inorganic Model (AIM) using a $SO_4^{2-}$-$NO_3^-$-$NH_4^+$-$H^+$ system (AIM-II) were
employed (Li et al., 2013).





**3 Results and discussion**
**3.1 General description of chemical components in Mt. Tai**

The concentrations of dicarboxylic acids and related SOA, EC, OC, WSOC and

inorganic ions in $PM_{2.5}$ samples from Mt. Tai are summarized in Table 1. During the
campaign the height of boundary layer at Mt. Tai was frequently reduced to ~600 m at night,
which kept the sampling site in the free troposphere at night. In contrast, the boundary layer
extended far above the mountaintop during the daytime (Zhu et al., 2018). However, as a
tracer of combustion source, EC concentration is very low and shows a similar level in the
day and night periods, suggesting that the impact of anthropogenic emissions from the
lowland region on the mountaintop atmosphere is insignificant. OC and WSOC in the $PM_{2.5}$
samples are higher in daytime than in nighttime (Table 1), largely due to the stronger
photochemical oxidation in daytime rather than the changes in the boundary layer heights.
OC/EC and WSOC/OC ratios are around 1.4 times higher in daytime than in nighttime (Fig.
4), indicating an enhancing SOA production due to the daytime photochemical oxidation
(Hegde and Kawamura, 2012).

The meteorological conditions in daytime can promote the secondary aerosol formation,

resulting in higher concentration of $SO_4^{2-}$ (13 ± 6.9μg m$^{-3}$, Table 1) compared to that (9.6 ±
3.7μg m$^{-3}$) in nighttime (Kundu et al., 2010a). On the contrary, $NO_3^-$ concentration is higher
in the nighttime (4.2 ± 2.3μg m$^{-3}$, Table 1) than in the daytime (3.0 ± 2.1μg m$^{-3}$), which is
probably caused by the evaporative loss of $NH_4NO_3$ due to the higher daytime temperature
(Table 1) (Pathak et al., 2009). Consequently, the concentrations of $NH_4^+$ also present higher
values in nighttime (6.6 ± 2.5μg m$^{-3}$, Table 1) compared to that in daytime (5.3 ± 2.9μg m$^{-3}$).
As shown in Table 1, the remaining four kinds of cations ($K^+$, $Na^+$, $Ca^{2+}$ and $Mg^{2+}$), which
can be regarded as the key markers of primary sources, did not exhibit significant diurnal
variations, again suggesting that the effect of boundary layer heights is minor. In this work,
LWC and $pH_{IS}$ are calculated by using AIM-II model, because both species cannot be
directly measured. LWC exhibits higher concentrations (94± 100μg m$^{-3}$) in daytime than
that (75 ± 69μg m$^{-3}$) in nighttime (Table 1). In contrast, $pH_{IS}$ shows lower values (-0.04 ± 0.5)



in daytime compared to that (0.4 ± 0.6) in nighttime (Table 1), indicating the daytime
aerosols are more acidic.

**3.2 Molecular distributions of dicarboxylic acids and related SOA**

A homologous series of dicarboxylic acids ($C_2$–$C_{11}$), ketocarboxylic acids ($\omega C_2$–$\omega C_9$
and pyruvic acid), and α-dicarbonyls (glyoxal and methylglyoxal) in $PM_{2.5}$ samples of Mt.
Tai were determined (Table 2). The molecular compositions of these compounds are
illustrated in Fig. 2.
Total dicarboxylic acids are 430 ± 282ng m$^{-3}$ (27–944 ng m$^{-3}$, Table 2) in daytime,
around two times higher than those in nighttime (282 ± 161 ng m$^{-3}$, 73–671ng m$^{-3}$). The
average concentration levels (354 ± 239 ng m$^{-3}$) are lower than those in Asian lowland (e.g.
14 Chinese cities (892 ± 457 ng m$^{-3}$) (Ho et al., 2007); Chennai in India (502.9 ± 117.9 ng
m$^{-3}$) (Pavuluri et al., 2010)) and elevated regions (e.g. Mt. Hua in central China (744 ± 340
ng m$^{-3}$) (Meng et al., 2014); the central Himalayan in Nainital, India (430 ng m$^{-3}$) (Hegde
and Kawamura, 2012)), but higher than those in the continental background area such as
Qinghai Lake in Tibetan Plateau (231 ± 119 ng m$^{-3}$) (Meng et al., 2013) and marine regions
such as North Pacific (68 ng m$^{-3}$) (Hoque et al., 2017) and the western North Pacific (99.2 ±
86.4 ng m$^{-3}$) (Boreddy et al., 2017).
Interestingly, we found that the levels of dicarboxylic acids are equivalent to those at
Mt. Fuji in Japan in day and night (day: 424 ng m$^{-3}$; night: 266 ng m$^{-3}$) (Mochizuki et al.,
2017), which are dominantly derived from the oxidation of biogenic VOCs such as isoprene
and α-pinene in summer. Both mountains are located at the similar latitude in East Asia, and
the altitudes of the sampling sites at Mt. Tai and Mt. Fuji are almost the same. Thus, one
may expect that the emissions of biogenic VOCs at both sites during the same season are
similar. Moreover, $O_3$ level during the observation period in Mt. Tai is also similar to that at
Mt. Fuji, Japan ranging from a few ppb at night to about 60 ppb (Mochizuki et al., 2017) at
the noontime, which means that photochemical activity at both sites during the campaigns
are similar. Therefore, concentrations of dicarboxylic acids are comparable at both sites with
a similar diurnal difference.



At the Mt. Tai site, the concentrations of dicarboxylic acids in daytime were about two
times higher than in nighttime, which can be ascribed to the stronger photochemical
production of dicarboxylic acids and/or higher emissions of the precursors in daytime. As
shown in Fig. 2, oxalic acid ($C_2$) is the dominant species in Mt. Tai, followed by malonic
acid ($C_3$), succinic acid ($C_4$), and azelaic acid ($C_9$) during the day and night, respectively.
These four species account for 60, 12, 7.2, and 6.9% of the total dicarboxylic acids in the
daytime and 53, 11, 8.5, and 7.6 % of the total in the nighttime, respectively. The molecular
compositions in Mt. Tai is similar to that in other remote areas such as Mt. Fuji, Japan, Mt.
Hua and Qinghai Lake, China in the summer (Meng et al., 2013, 2014; Mochizuki et al.,
2017), but different from that in urban regions where phthalic and/or tere-phthalic acids are
more abundant than $C_9$ because of higher emissions of anthropogenic precursors (e.g.,
aromatics and plasticizers) (Cheng et al., 2015; He et al., 2014; Jung et al., 2010; Wang et al.,

2002, 2017).

Ketocarboxylic acids are the major intermediates of aqueous phase photochemical
oxidation producing dicarboxylic acids in the atmosphere (Kawamura and Ikushima, 1993;
Pavuluri and Kawamura, 2016), which are $43 \pm 28$ ng m$^{-3}$ in the daytime and $37 \pm 19$ ng m$^{-3}$
in the nighttime, respectively, with glyoxylic acid ($\omega C_2$) being the dominant $\omega$-oxoacid,
followed by pyruvic acid (Pyr) and 3-oxobutanoic acid ($\omega C_3$) (Table 2 and Fig. 2). Previous
studies have proposed that $\omega C_2$ can be initially formed from photochemical oxidation of
glyoxal with OH radical and other oxidants in aqueous phase and then further oxidized into
oxalic acid (Rapf et al., 2017; Wang et al., 2012). In contrast to the diurnal variations of
dicarboxylic and ketocarboxylic acids, the concentrations of $\alpha$-dicarbonyls exhibit higher
concentrations in nighttime than those in daytime (Fig. 2). Because $\alpha$-dicarbonyls are the
major precursors produced from the photochemical oxidation of isoprene and other VOCs in
the atmosphere (Carlton et al., 2006, 2007), the opposite pattern suggests that the aerosol
aqueous phase oxidation in nighttime is impressed in comparison with that in daytime. The
concentrations of Gly are less than mGly, largely because of the stronger biogenic sources





and the lower oxidation rate of mGly with OH radical in aerosol phase compared to Gly
(Cheng et al., 2013; Meng et al., 2013).
Temporal variations in concentrations of total dicarboxylic acids, ketocarboxylic acids
and α-dicarbonyls are illustrated in Fig. 3, along with the meteorological parameters. During
the whole sampling periods, the concentrations of total daicarboxylic acids and related SOA
fluctuated significantly with a maximum (1060 ng m$^{-3}$) on August 4 and a minimum (33 ng
m$^{-3}$) on August 7. Our results showed that the levels of water-soluble organic compounds
decrease by 30-80% when it was rainy, suggesting that dicarboxylic acids and related SOA
can be removed efficiently by the wet precipitation, because these water-soluble compounds
are not only easily washed out but also can be efficiently removed by serving as cloud
condensation nuclei (CCN) during the wet deposition (Leaitch et al., 1996). Moreover, a
reduced secondary formation due to weaker solar radiation and a reduced biogenic emission
during the rainy days are also responsible for the lowest concentrations of dicarboxylic acid
and related SOA.
**3.3 Biogenic versus anthropogenic and local versus long-range transport sources**
Previous studies have proposed that $C_2$, $C_3$ and $C_4$ are produced by the photochemical
degradation of longer-chain diacids, while $C_3$ is produced by photooxidation of $C_4$ in the
atmosphere (Hoque et al., 2017; Kawamura and Usukura, 1993; Kunwar et al., 2017).
Therefore, both ratios of $C_2/C_4$ and $C_3/C_4$ can be regarded as indicators of photochemical
aging of organic aerosols. The $C_2/C_4$ and $C_3/C_4$ ratios in the mountainous atmosphere are 8.0
$\pm$ 2.7 and 1.6 $\pm$ 0.6, respectively, higher than aerosols freshly emitted from sources such as
vehicle exhausts ($C_2/C_4$: 7.1; $C_3/C_4$: 1.3) (Ho et al., 2006) and biomass burning plumes
($C_2/C_4$: 5.0; $C_3/C_4$: 0.7) (Kundu et al., 2010b), but lower than photochemically aged aerosols
in remote regions such as a continental background site in Tibet Plateau ($C_2/C_4$: 11±7.2;
$C_3/C_4$: 2.2±1.3) (Meng et al., 2013) and the North and South Pacific ($C_2/C_4$: 8.7; $C_3/C_4$:
3.0)(Hoque et al., 2017). Compared with those in the nighttime, the higher ratios of $C_2/C_4$
and $C_3/C_4$ (Fig. 4) in the daytime again indicate that the photochemical modification of
aerosols is stronger. A few studies have found that when local sources are dominant over



long-range transport, both ratios of $C_2/C_4$ and $C_3/C_4$ would correlate strongly with the
ambient temperatures (Meng et al., 2013; Pavuluri et al., 2010). In the current work, the
ratios of $C_2/C_4$ ($R^2$>0.5) (Fig. 5a) and $C_3/C_4$ ($R^2$≥0.5) (Fig. 5b) correlated well with the
ambient temperatures in both the daytime and the nighttime, clearly suggesting that
dicarboxylic acids and related SOA at the Mt. Tai during the campaign are mostly derived
from the local photochemical oxidation of BVOCs rather than long-range transport.
Aggarwal et al., (2008) found that diacid-C/OC and $C_2$/total diacids should increase in
daytime when local emission and photooxidation are more significant than long-range
transport. In the summit of Mt. Tai the daytime ratios of diacid-C/OC and $C_2$/total diacids
are 5.5 ± 2.6% and 60 ± 7.7%, which are about 1.2 and 1.3 times higher than those in the
nighttime, respectively (Fig. 4), further indicating the stronger photochemical oxidation in
daytime and the dominance of local sources for the SOA production in the troposphere of
Mt. Tai.

Both ratios of $C_9/C_6$ and $C_9$/Ph can be used as indicators to qualitatively evaluate the

source strength of anthropogenic versus biogenic precursors for producing dicarboxylic
acids and related SOA (Jung et al., 2010), because $C_6$ and Ph are largely produced by the
oxidation of anthropogenic cyclohexene (Hatakeyama et al., 1987) and aromatic
hydrocarbons such as naphthalene (Kawamura and Ikushima, 1993), respectively. In
contrast, $C_9$ is mainly produced by the oxidation of biogenic unsaturated fatty acids
containing a double bond at the C-9 position (Wang et al., 2010). As shown in Fig. 4, both
ratios of $C_9/C_6$ and $C_9$/Ph are higher in the daytime than those in the nighttime, which is
mainly attributed to the stronger biogenic activity of vegetation in daytime in the mountain
regions. The average values of $C_9/C_6$ (14±9.0) and $C_9$/Ph (7.2±2.2) at the mountaintop of Mt.
Tai are higher than those in urban regions such as Xi'an China ($C_9/C_6$: 3.1; $C_9$/Ph:
5.6)(Cheng et al., 2013) and also higher than other mountainous in summer such as Mt.
Himalayan, India ($C_9/C_6$: 2.1; $C_9$/Ph: 0.2)(Hedge and Kawamura, 2012) and Mt. Fuji, Japan
($C_9/C_6$: 3.1) (Mochizuki et al., 2017). Model simulation (Fu et al., 2008) and field
observations (Meng et al., 2014) have suggested that the concentration ratio of particulate



Gly/mGly is about 1:5 when biogenic sources are predominant and is about 1:1 when
anthropogenic sources are predominant such as in urban areas. As shown in Table 3, the
ratios of Gly/mGly in the Mt. Tai atmosphere are 1:5.1 in daytime and 1:4.8 in nighttime,
further suggesting that Gly and mGly in the Mt. Tai samples are mostly derived from
biogenic sources. This result is in agreement with the high abundance of $C_9$ relative to the
total dicarboxylic acids (7.2%), which is about two times higher than that (3.5%) in 14
Chinese megacities in the summer (Ho et al., 2007). Moreover, a trace amount of elemental
carbon (EC) was found for most of the samples (Table 1), suggesting that the impact of
pollutants derived from anthropogenic sources on the mountaintop atmosphere during the
campaign are negligible. Consequently, it can be concluded that the summertime SOA of Mt.
Tai are mainly derived from local photochemical oxidation of biogenic precursors rather
than long-range transport of anthropogenic precursors during the sampling period.
**3.4 Production of dicarboxylic acids and related SOA from biogenic sources**
A three-dimensional modeling study has proposed that 79% of oxalic acid is originated
from the photochemical oxidation of isoprene and other biogenic hydrocarbons in cloud
(Myriokefalitakis et al., 2011). Laboratory experiments and model simulations have
demonstrated that the photooxidaltaion of isoprene (Carlton et al., 2006, 2007; Huang et al.,
2011) and monoterpenes (Fick et al., 2003; Lee et al., 2006) can produce Gly and mGly via
reactions with OH radical and/or $O_3$ in the aerosol aqueous phase or the gas phase and
subsequently partition into cloud droplets, where both carbonyls are oxidized further by OH
radical to form oxalic acid (Lim et al., 2005; Tan et al., 2010).
In order to further ascertain the contribution of BVOCs to dicarboxylic acids and
related SOA during the high biological activity period in Mt. Tai, SOA tracers derived from
isoprene-, α-/β-pinene- and β-caryophyllene in the $PM_{2.5}$ samples collected at the Mt. Tai
site were determined. Their total concentrations (the sum of isoprene+ pinene+
caryophyllene derived SOA tracers) are 1.3 times higher in the daytime ($106 \pm 56$ ng m$^{-3}$)
than those in the nighttime ($79 \pm 38$ ng m$^{-3}$) (unpublished data), which is consistent with the
diurnal variation patterns of dicarboxylic acids, ketocarboxylic acids and WSOC (Tables 1



and 2). Previous studies reported that 2-methylglyceric acid, which is an isoprene oxidation
product, and 3-hydroxyglutaric acid, which is α-/β-pinene oxidation product, can serve as
organic precursors for the production of dicarboxylic acids and ketocarboxylic acids (Fu et
al., 2013). As shown in Table 4, major dicarboxylic acids and related SOA (e.g. $C_2$, $C_2$, Gly
and mGly) correlated positively with isoprene oxidation products during daytime and
nighttime ($R$>0.60, $P$<0.01), indicating that isoprene oxidation products can serve as
precursors for the production of oxalic acid via α-dicaronyls oxidation (Myriokefalitakis et
al., 2011). Moreover, both α-/β-pinene and caryophyllene oxidation products also presented
strong correlations with dicarboxylic acids and related SOA ($R$>0.55, $P$<0.01) (Table 4),
further highlighting the important contribution of BSOA to dicarboxylic acids and related
SOA in Mt. Tai during summer.
**3.5 Effects of temperature, relative humidity, and $O_3$ concentrations on the formation**
**of oxalic acid and related SOA**

Because oxidants such as OH radicals were not measured in Mt. Tai, $O_3$ is considered

here as an indicator of the total oxidant concentrations in this study. A significant linear
correlation of oxalic acid with $O_3$ concentrations is observed for the daytime samples
($R^2$=0.91), but no correlation ($R^2$=0.05) was found for the nighttime samples (Fig.6a). Such a
phenomenon was also observed in Mt. Fuji, Japan (Mochizuki et al., 2017) and Beijing,
China (He et al., 2014). Additionally, $C_2$/ Gly, $C_2$/ mGly and $C_2$/total diacids ratios correlate
positively with $O_3$ concentrations in the daytime, but such correlations were not found in the
nighttime (Fig 8(a-c)). Mochizuki et al. (2017) have reported a robust correlation between
concentration ratios of oxalic acid to isoprene plus α-pinene (oxalic
acid/(isoprene+α-pinene)) and $O_3$ concentrations in a large forest region of Mt. Fuji, Japan
in the daytime. In the curren work, BSOA tracers correlate strongly with $O_3$ concentrations
in the daytime ($R$>0.6, $P$<0.01), but no correlation was found at night (Table 4). These
results suggest that the daytime oxalic acid and related SOA in the mountaintop of Mt. Tai
are largely derived from $O_3$ oxidation of BVOCs such as isoprene and α-pinene, while the





nighttime oxalic acid and related SOA might be primarily produced by $NO_3$ radical and
other oxidizing agents such as $H_2O_2$ (Claeys et al., 2004; Herrmann et al., 1999).

As shown in Table 4, nearly all of the detected BSOA tracers including

2-methylglyceric acid, 3-hydorxyglutaric acid and β-caryophyllinic acid exhibit a strong
correlation with the ambient temperature, largely due to the increased production of BSOA
from enhanced emissions of BVOCs under the higher temperature conditions. The BSOA
tracer concentrations are higher in daytime than in nighttime, largely due to more abundant
BVOC emissions because of stronger biogenic activity of vegetation during the daytime. In
addition, oxalic acid and $C_2$/total diacids ratios correlated strongly with temperatures (Fig.6b
and Fig.8f), because higher temperature conditions can promote photochemical formation of
oxalic acid. Such a temperature dependence is also observed in other regions such as Mt.
Hua (Meng et al., 2014) and Beijing (Wang et al., 2017) in China.

Online measurements, field observations and chamber studies (Cheng et al., 2017; Gao

et al., 2004; McNeill, 2015; Meng et al., 2014; Wang et al., 2012, 2017) have suggested that
oxalic acid is primarily derived from the acid-catalyzed heterogeneous oxidation of glyoxal
and related precursors in the aqueous phase. Here we investigate the impact of LWC and
$pH_{IS}$ on the formation of oxalic acid in Mt. Tai aerosols. As shown in Fig.6c, a strong linger
correlation between $C_2$ and $SO_4^{2-}$ was found for the daytime ($R^2$=0.89) and nighttime
($R^2$=0.76) samples, respectively, which is consistent with the measurements observed in
other mountainous region (Meng et al., 2014) and Chinese cities (Wang et al., 2002, 2010,
2012, 2017; Yu et al., 2005), indicating that oxalic acid and sulfate are formed via a similar
formation pathway such as in-cloud or aqueous-phase (Warneck, 2003). In this study, oxalic
acid does not exhibit correlations with relative humidity (RH) and LWC (Fig.6d and 6e), but
presents a significant negative correlation with $pH_{IS}$ ($R^2$>0.60) (Fig.6f), largely due to the
fact that acidic conditions can promote the formation of oxalic acid and their precursors.
Therefore, a robust negative correlation was obtained for $pH_{IS}$ and the precursors of oxalic
acid such as Gly, mGly and $\omega C_2$ ($R^2$>0.50). A few studies have pointed out that aerosol
acidity are favorable for the formation of biogenic SOA (BSOA) derived from isoprene



oxidation such as 2-methylglyceric acid, which can be oxidized into Gly and mGly and then
converted to oxalic acid (Meng et al., 2014; Surratt et al., 2007, 2010). Our previous studies
have revealed that enhanced RH can reduce particle acidity (pH$_{IS}$) and is thus unfavorable
for oxalic acid formation by acid-catalyzed reactions occurring in the aerosol aqueous phase
(Li et al., 2013, 2018; Meng et al., 2014). RH is a key factor controlling the aerosol LWC
(Bikkina et al., 2017). Deshmukh et al. (2017) and Bikkina et al. (2017) also found that RH
and LWC correlated well with oxalic acid, indicating that humid conditions are favorable for
the aqueous phase formation of C$_2$. Nevertheless, Zhang et al. (2011) pointed out that low
RH conditions can promote SOA yields via the oxidation of isoprene. Higher RH and LWC
can promote the partitioning of water-soluble semivolatile organic precursors of oxalic acid
(e.g., Gly and mGly) into the aerosol aqueous phase but can also suppress acid-catalyzed
formation of oxalic acid because of lower aerosol acidity due to dilution. Therefore, C$_2$ does
not present any correlations with RH or LWC in Mt. Tai.
**3.6 Stable carbon isotopic composition of oxalic acid and related SOA**
To further understand the formation mechanism of C$_2$ and related SOA, the stable
isotopic composition of major dicarboxylic acids and related SOA in the Mt. Tai aerosols
were investigated (Table 5). Generally, an increase in δ$^{13}$C values was observed with a
decrease in carbon numbers of dicarboxylic acids. The averaged δ$^{13}$C value (daytime:
−15.8±1.9‰; nighttime: −17.2±1.7‰) of C$_2$ is higher than other dicarboxylic acid and
related SOA in the Mt. Tai atmosphere, and also higher than those observed in the urban
regions such as Xi'an, China (−22.7‰ to −22.0‰) (Wang et al., 2012) and Sapporo, Japan
(18.8±2.0‰) (Aggarwal and Kawamura, 2008) and the rural regions such as Morogoro,
Tanzania (18.3±1.7‰) (Mkoma et al., 2014), but lower than those (11.5±2.8‰) (Zhang et
al., 2016) at a background supersite (the Korea Climate Observatory at Gosan) in East Asia
during the summer. Pavuluri et al. (2016) have reported that the average δ$^{13}$C values of C$_2$
from biogenic aerosols are higher than those from anthropogenic aerosols. The relatively
higher δ$^{13}$C values of C$_2$ observed in Mt. Tai further demonstrate that the contribution of
biogenic sources to C$_2$ and related SOA is more significant than anthropogenic sources,



which is consistent with our discussion above. The average $\delta^{13}C$ values of $C_4$ are more
negative than $C_2$ and $C_3$ (Fig. 7). Such a phenomenon is also observed in other regions
(Aggarwal and Kawamura, 2008;Wang et al., 2012; Zhang et al., 2016). Photochemical
decomposition (or breakdown) of longer-chain diacarboxylic acids (e.g. $C_3$ or $C_4$) in aerosol
aqueous phase can form $C_2$ (Wang et al., 2017), during which $C_3$ or $C_4$ release $CO_2/CO$ by
reaction with OH radical and other oxidants, resulting in $C_2$ more enriched in $^{13}C$ due to
kinetic isotope effects (KIE) (Wang et al., 2012). The $^{13}C$ enrichment in $C_2$ is more
distinguished in daytime than in nighttime (Table 5 and Fig. 7), largely due to the enhanced
photochemical oxidation. However, such diurnal variation was not found for $C_3$ and $C_4$.

$\omega C_2$ is an important intermediate of aqueous phase photochemical oxidation of

precursors such as Gly, mGly, and Pyr during the $C_2$ formation process (Carlton et al., 2006;
Fu et al., 2008; Wang et al., 2012). Thus, the mass ratios of $C_2/\omega C_2$, $C_2/Gly$ and $C_2/mGly$ are
indicative of organic aerosol aging (Wang et al., 2017). As shown in Fig. 8(g-i), $\delta^{13}C$ values
of $C_2$ correlate robustly with $C_2/Gly$, $C_2/mGly$, and $C_2$/total diacids, suggesting an
enrichment of $^{13}C$ during the organic aerosol ageing process. During the campaign, $\omega C_2$ is
less enriched in $^{13}C$ in comparison with Gly, mGly, and Pyr, because lighter isotope ($^{12}C$) is
preferentially enriched in the products due to KIEs during the aqueous phase irreversible
chemical reactions (Wang et al., 2012). As one of the major precursors of Gly, isoprene
emitted directly from vegetation is depleted in $^{13}C$ with a range from –32‰ to –27‰ (Affek
and Yakir, 2003), but during the transport process isoprene could gradually be enriched with
$^{13}C$ ($\delta^{13}C$ value=–16.8‰) due to isotope fractionation associated with the reaction with OH
radical (Rudolph et al., 2003). Moreover, chamber experiments have pointed out that
$\beta$-pinene is preferably enriched with $^{13}C$ during its ozonolysis due to KIE (Fisseha et al.,
2009). Therefore, the relatively higher values of Gly and mGly can be attributed to the
secondary formation from the oxidation of isoprene and other biogenic precursors with
stronger enrichment of $^{13}C$.
**4. Summary and conclusions**





PM$_{2.5}$ aerosols from the summit of Mt. Tai (15340 m a.s.l) in the North China Plain
during the summer of 2016 were analyzed for dicarboxylic acids, ketocarboxylic acids,
α-dicarbonyls, EC, OC and WSOC. Molecular compositions of dicarboxylic acids and
related compounds in the forested highland are similar to those on the ground surface and
other mountainous regions. The concentrations of total dicarboxylic acids and
ketocarboxylic acids are higher in daytime than those in daytime, but α-dicarbonyls presents
lower values in daytime, suggesting the mountainous atmospheric environment is more
photochemically aged in daytime than in nighttime. The concentrations of oxalic acid and
BSOA traces and the mass ratios of C$_2$/Gly, C$_2$/mGly, and C$_2$/total diacids correlate
positively with O$_3$ concentrations in the daytime during the campaign, but such correlations
were not found at night, suggesting that in the mountaintop atmosphere O$_3$ oxidation is the
major formation pathway of oxalic acid and related SOA in daytime. Moreover, C$_2$, C$_2$/total
diacids ratios and BSOA tracers correlate strongly with temperatures, because higher
temperature conditions can enhance the emissions of BVOCs and further promote the
photochemical formation of C$_2$. C$_2$ has a robust correlation with pH$_{IS}$ and SO$_4^{2-}$ during the
whole sampling period, indicating that acidic conditions can favor the formation of oxalic
acid in aqueous phase.
A significant enrichment of $^{13}$C in dicarboxylic aicds was observed as a function of
their carbon number. The observed larger δ$^{13}$C values of lower carbon numbered
dicarboxylic acids can be explained by isotopic fractionations resulting from the
atmospheric decomposition of relatively longer chain-diacids or their precursors. Increased
δ$^{13}$C values of C$_2$ relative to C$_2$/Gly and C$_2$/mGly ratios also suggested an important effect
of photochemical aging on the stable carbon isotopic composition of dicarboxylic acids.

**Acknowledgements**
This work was supported by China National Science Funds (Grant No. 41505112 and
41773117), Natural Science Foundation of Shandong Province (Grant No. BS2015HZ002),
the China National Natural Science Funds for Distinguished Young Scholars (No.41325014)
and Open Funds of State Key Laboratory of Loess and Quaternary Geology, Institute of



Earth Environment, Chinese Academy of Sciences (Grant Nos. SKLLQG1509 and
SKLLQG1504).

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





Table 1. Meteorological parameters and concentrations of inorganic ions, ozone, water
soluble organic carbon (WSOC), organic carbon (OC), elemental carbon (EC), liquid water
content (LWC), and in-situ pH ($pH_{IS}$) of $PM_{2.5}$ from Mt. Tai in the summer.

| | Daytime (*n*=28) | Nighttime (*n*=29) | Total (*n*=57) |
|---|---|---|---|
| **I. Meteorological parameters and ozone** | | | |
| Temperature (°C) | 23 ± 2.9 (17–28) | 19 ± 2.9 (12–25) | 21 ± 3.6 (12–28) |
| Relative humidity (%) | 92 ± 5.0 (80–98) | 77 ± 8.2 (65–193) | 84 ± 9.9 (65–98) |
| $O_3$ (ppb) | 32 ± 16 (7.8–61) | 22 ± 12 (6.0–48) | 27 ± 15 (6.0–61) |
| **II. Inorganic ions (µg m$^{-3}$)** | | | |
| $K^+$ | 0.4 ± 0.2 (0.1–0.8) | 0.4 ± 0.2 (0.1–0.7) | 0.4 ± 0.2 (0.1–0.8) |
| $Na^+$ | 0.3 ± 0.1 (0.1–0.9) | 0.3 ± 0.2 (0.1–1.0) | 0.3 ± 0.2 (0.1–1.0) |
| $NH_4^+$ | 5.3 ± 2.9 (0.5–12) | 6.6 ± 2.5 (1.2–11) | 5.9 ± 2.8 (0.5–12) |
| $Mg^{2+}$ | 0.2 ± 0.1 (0–0.3) | 0.2 ± 0.1 (0.1–0.3) | 0.2 ± 0.1 (0–0.3) |
| $Ca^{2+}$ | 0.3 ± 0.1 (0–0.5) | 0.3 ± 0.2 (0.1–0.7) | 0.3 ± 0.2 (0–0.7) |
| $NO_3^-$ | 3.0 ± 2.1 (0.1–8.4) | 4.2 ± 2.3 (0.9–10) | 3.6 ± 2.3 (0.1–10) |
| $SO_4^{2-}$ | 13 ± 6.9 (3.0–33) | 9.6 ± 3.7 (2.9–18) | 12 ± 5.8 (2.9–33) |
| Subtotal | 23 ± 12 (4.6–54) | 22 ± 8.2 (6.6–40) | 22 ± 10 (4.6–54) |
| **Ⅲ. Other species (µg m$^{-3}$)** | | | |
| EC | 0.2 ± 0.2 (0–0.6) | 0.2 ± 0.2 (0–0.8) | 0.2 ± 0.2 (0–0.8) |
| OC | 2.4 ± 0.8 (1.1–3.9) | 2.1 ± 0.3 (1.5–2.8) | 2.2 ± 0.6 (1.1–3.9) |
| WSOC | 1.9 ± 0.8 (0.8–3.6) | 1.4 ± 0.5 (0.7–2.3) | 1.7 ± 0.7 (0.7–3.6) |
| $pH_{IS}$ | -0.04 ± 0.5 (-0.9–1.0) | 0.4 ± 0.6 (-1.0–1.2) | 0.2 ± 0.6 (-1.0–1.2) |
| LWC | 94 ± 100 (10–313) | 75 ± 69 (6.3–199) | 84 ± 86 (6.3–313) |
| $PM_{2.5}$ | 38 ± 19 (6.1–83) | 36 ± 14 (11–66) | 37 ± 16 (6.1–83) |




Table 2. Concentrations (ng m$^{-3}$) of dicarboxylic acids, ketocarboxylic acids and
α-dicarbonyls of PM$_{2.5}$ from Mt. Tai in the summer.

| Compounds | Daytime ($n=28$) | Nighttime ($n=29$) | Total ($n=57$) |
|---|---|---|---|
| **I. Dicarboxylic acids** | | | |
| Oxalic, C$_2$ | 272± 190 (11–623) | 156 ± 105 (34–415) | 213 ± 162 (11–623) |
| Malonic,C$_3$ | 49 ± 30 (4.0–101) | 31 ±17 (7.4–69) | 40 ± 26 (4.0–101) |
| Succinic, C$_4$ | 30 ± 23 (2.0–83) | 24 ± 16 (4.7–67) | 27 ± 20 (2.0–83) |
| Glutaric, C$_5$ | 7.0 ± 5.5 (0.4–19) | 5.6 ± 3.9 (1.1–14) | 6.3 ± 4.8 (0.4–19) |
| Adipic, C$_6$ | 2.2 ± 1.7 (0.1–5.6) | 2.2 ± 1.8 (0.2–7.7) | 2.2 ± 1.7 (0.1–7.7) |
| Pimelic, C$_7$ | 3.0 ±1.9 (0.3–7.3) | 2.9 ± 1.3 (0.3–6.1) | 3.0 ± 1.6 (0.3–7.3) |
| Suberic, C$_8$ | 4.3 ± 2.2 (0.9–9.0) | 3.8 ± 2.8 (0.4–13) | 4.0 ± 2.5 (0.4–13) |
| Azelaic, C$_9$ | 24 ± 14 (4.2–55) | 19 ± 8.6 (4.5–41) | 22 ± 12 (4.2–55) |
| Sebacic, C$_{10}$ | 5.9 ± 4.3 (0.1–14) | 5.6 ± 2.7 (0.7–11) | 5.8 ± 3.6 (0.1–14) |
| Undecanedioic, C$_{11}$ | 2.4 ± 1.7 (0.2–5.8) | 1.1 ± 0.8 (0–3.8) | 1.7 ± 1.4 (0–5.8) |
| Methylmalonic, iC$_4$ | 2.1 ±1.7 (0.1–5.2) | 2.1 ± 1.5 (0–5.3) | 2.1 ± 1.6 (0–5.3) |
| Mehtylsuccinic, iC$_5$ | 2.7 ± 2.0 (0.1–7.1) | 2.2 ± 1.7 (0.2–6.1) | 2.4 ± 1.8 (0.1–7.1) |
| Methyglutaric, iC$_6$ | 2.6 ± 2.1 (0.5–9.1) | 2.3 ± 1.9 (0–9.0) | 2.5 ± 2.0 (0–9.1) |
| Maleic, M | 2.0 ±1.2 (0.1–4.3) | 3.0 ± 2.0 (0.7–8.2) | 2.5 ± 1.7 (0.1–8.2) |
| Fumaric, F | 4.2 ± 2.7 (0.2–9.4) | 4.0 ± 3.0 (0.5–13) | 4.1 ± 2.8 (0.2–13) |
| Methylmaleic, mM | 2.9 ± 1.7 (0.1–6.6) | 2.7 ± 2.1 (0.5–9.9) | 2.8 ± 1.9 (0.1–9.9) |
| Phthalic, Ph | 3.0 ± 1.5 (0.6–5.6) | 3.3 ± 2.3 (0.7–11.2) | 3.2 ± 1.9 (0.6–11.2) |
| Isophthalic, iPh | 1.6 ± 1.0 (0.1–3.3) | 1.3 ± 0.8 (0.2–3.5) | 1.4 ± 0.9 (0.1–3.5) |
| Terephthalic, tPh | 1.9 ± 1.3 (0.1–5.0) | 2.4 ±1.5 (0.1–6.1) | 2.2 ± 1.4 (0.1–6.1) |
| Ketomalonic, kC$_3$ | 2.6 ± 1.5 (0–5.8) | 2.7 ± 1.5 (0.5–6.4) | 2.7 ± 1.5 (0–6.4) |
| Ketopimelic, kC$_7$ | 3.6 ± 2.8 (0.2–9.3) | 3.9 ± 2.6 (0.2–12) | 3.7 ± 2.7 (0.2–12) |
| Subtotal | 430 ± 282 (27–944) | 282 ± 161 (73–671) | 354 ± 239 (27–944) |
| **II. Ketocarboxylic acids** | | | |
| Pyruvic, Pyr | 14 ± 8.8 (1.4–28) | 11 ± 5.5 (2.2–23) | 12 ± 7.4 (1.4–28) |
| Glyoxylic, ωC$_2$ | 18 ± 12 (0.9–38) | 15 ± 9.5 (3.5–35) | 16 ± 11(0.9–38) |
| 3-Oxopropanoic, ωC$_3$ | 4.0 ± 2.7 (0.1–7.7) | 4.1 ± 2.2 (0.5–8.3) | 4.1 ± 2.4 (0.1–8.3) |
| 4-Oxobutanoic, ωC$_4$ | 2.9 ± 1.9 (0.2–6.8) | 2.5 ± 1.7 (0.6–7.1) | 2.7 ± 1.8 (0.2–7.1) |
| 7-Oxoheptanoic, ωC7 | 1.0 ±0.6 (0–2.7) | 1.3 ± 1.0 (0.1–4.8) | 1.2 ± 0.9 (0.0–4.8) |
| 8-Oxooctanoic, ωC$_8$ | 1.5 ± 0.9 (0.1–3.3) | 1.5 ± 0.7 (0.2–3.4) | 1.5 ± 0.8 (0.1–3.4) |
| 9-Oxononanoic, ωC$_9$ | 2.0 ± 1.4 (0.1–4.4) | 1.8 ± 1.1 (0.2–4.3) | 1.9 ± 1.3 (0.1–4.4) |
| Subtotal | 43 ± 28 (2.9–88) | 37 ± 19 (7.6–77) | 40 ± 24 (2.9–88) |
| **III. α-Dicarbonyls** | | | |
| Glyoxal, Gly | 3.1 ± 1.8 (0.3–6.0) | 4.6 ± 2.6 (0.4–12) | 3.8 ± 2.3 (0.3–12) |
| Methyglyoxal, mGly | 16 ± 9.5 (1.8–33) | 22 ± 15 (1.4–62) | 19 ± 13 (1.4–62) |
| Subtotal | 19 ± 11 (2.6–39) | 27 ± 17 (2.1–69) | 23 ± 15 (2.1–69) |
| Total detected | 491 ± 320 (33–1060) | 346 ± 194 (96–807) | 417 ± 271 (33–1060) |





Table 3. Concentrations of α-dicarbonyls in PM$_{2.5}$ from Mt. Tai and Mt. Hua in China and
the global budgets of atmospheric Gly and mGly.

| Site | Sources/Season | Abundance | | Mass ratio |
|---|---|---|---|---|
| | | Gly | mGly | Gly/mGly |
| Global budget (Tg a$^{-1}$) | Biogenic | 22.8[a] | 113.5[a] | 1:5 |
| | Anthropogenic | 22.2[a] | 26.5[a] | 1:1 |
| Mt. Hua (ngm$^{-3}$) | Biogenic, Summer | 2.3[b] | 10[b] | 1:4.4 |
| | Anthropogenic, Winter | 8.8[b] | 1.3[b] | 1:1.5 |
| Mt. Tai (ngm$^{-3}$, this study) | Summer, Daytime | 3.1 | 15.8 | 1:5.1 |
| | Summer, Nighttime | 4.6 | 22.1 | 1:4.8 |

Note: [a] Data are calculated from Fu et al., 2008;
[b] Data are cited from Meng et al., 2014.

Table 4. Correlation coefficients (*R*) matrix among major low molecular weight dicarboxylic
acids and related SOA, BSOA tracers, temperature (T), and O$_3$ concentrations in Mt. Tai
during the summer campaign.

| | C$_2$ | C$_3$ | C$_4$ | ωC$_2$ | Pyr | Gly | mGly | O$_3$ | T |
|---|---|---|---|---|---|---|---|---|---|
| (a) Daytime | | | | | | | | | |
| 2-methylglyceric acid | 0.98[a] | 0.96[a] | 0.86[a] | 0.95[a] | 0.73[a] | 0.96[a] | 0.94[a] | 0.92[a] | 0.85[a] |
| 2-methylthreitol | 0.83[a] | 0.80[a] | 0.64[a] | 0.74[a] | 0.77[a] | 0.77[a] | 0.82[a] | 0.85[a] | 0.72[a] |
| 2-methylerthritol | 0.84[a] | 0.87[a] | 0.70[a] | 0.78[a] | 0.83[a] | 0.83[a] | 0.84[a] | 0.80[a] | 0.71[a] |
| *cis*-pinonic acid | 0.83[a] | 0.75[a] | 0.73[a] | 0.75[a] | 0.74[a] | 0.71[a] | 0.77[a] | 0.80[a] | 0.72[a] |
| 3-hydroxyglutaric acid | 0.81[a] | 0.76[a] | 0.69[a] | 0.74[a] | 0.78[a] | 0.74[a] | 0.73[a] | 0.73[a] | 0.75[a] |
| MBTCA | 0.84[a] | 0.77[a] | 0.83[a] | 0.82[a] | 0.75[a] | 0.74[a] | 0.77[a] | 0.82[a] | 0.67[a] |
| *β*-Caryophyllinic acid | 0.75[a] | 0.70[a] | 0.79[a] | 0.70[a] | 0.70[a] | 0.71[a] | 0.72[a] | 0.65[a] | 0.57[a] |
| (b) Nighttime | | | | | | | | | |
| 2-methylglyceric acid | 0.87[a] | 0.72[a] | 0.74[a] | 0.70[a] | 0.64[a] | 0.44[b] | 0.51[a] | 0.09 | 0.59[a] |
| 2-methylthreitol | 0.81[a] | 0.71[a] | 0.58[a] | 0.71[a] | 0.66[a] | 0.60[a] | 0.47[b] | 0.05[a] | 0.65[a] |
| 2-methylerthritol | 0.83[a] | 0.72[a] | 0.62[a] | 0.79[a] | 0.73[a] | 0.67[a] | 0.61[a] | 0.23 | 0.74[a] |
| *cis*-pinonic acid | 0.83[a] | 0.65[a] | 0.57[a] | 0.75[a] | 0.61[a] | 0.56[a] | 0.54[a] | 0.22 | 0.63[a] |
| 3-hydroxyglutaric acid | 0.79[a] | 0.62[a] | 0.69[a] | 0.71[a] | 0.60[a] | 0.58[a] | 0.50[a] | 0.43[b] | 0.62[a] |
| MBTCA[c] | 0.82[a] | 0.80[a] | 0.78[a] | 0.73[a] | 0.75[a] | 0.61[a] | 0.55[a] | 0.30 | 0.60[a] |
| *β*-caryophyllinic acid | 0.68[a] | 0.74[a] | 0.61[a] | 0.73[a] | 0.71[a] | 0.73[a] | 0.58[a] | 0.32[a] | 0.53[a] |

[a] *P*<0.01; [b] *P*<0.05.
[c] MBTCA: 3-Methyl-1,2,3-butanetricarboxylic acid.








Table 5. Stable carbon isotopic compositions ($\delta^{13}$C, ‰) of major dicarboxylic acids and
related SOA in PM$_{2.5}$ of Mt. Tai in the North China Plain.

| Compounds | Daytime ($n$=28) | Nighttime ($n$=29) | Total ($n$=57) |
|---|---|---|---|
| **I. Dicarboxylic acids** | | | |
| Oxalic, C$_2$ | −15.8±1.9 (−19.4 to −13.0) | −17.2±1.7 (−20.1 to −12.1) | −16.5±1.9 (−20.1 to −12.1) |
| Malonic, C$_3$ | −19.1±2.3 (−23.8 to−15.9) | −18.5±1.8 (−21.1 to−15.3) | −18.8±2.0 (−23.8 to−15.3) |
| Succinic, C$_4$ | −22.0±2.3 (−25.6 to−18.5) | −21.4±2.2 (−24.6 to −18.4) | −21.7±2.2 (−25.6 to −18.4) |
| Adipic, C$_6$ | −23.7±2.5 (−27.3 to −19.9) | −24.8±2.4 (−27.9 to −21.4) | −24.2±2.5 (−27.9 to −19.9) |
| Azelaic, C$_9$ | −24.7±2.6 (−28.7 to−21.0) | −25.7±2.7 (−30.3 to−21.9) | −25.2±2.7 (−30.3 to −21.0) |
| Phthalic, Ph | −24.3±2.5 (−28.1 to−20.6) | −25.2±2.6 (−29.2 to−20.9) | −24.8±2.5 (−29.2 to −20.6) |
| **II. Ketocarboxylic acids** | | | |
| Pyruvic, Pyr | −19.4±2.1 (−23.1 to −16.5) | −21.2±2.2 (−24.5 to −17.8) | −20.3±2.3 (−24.5 to −16.5) |
| Glyoxylic, ωC$_2$ | −18.6.8±1.9 (−21.5 to −15.6) | −20.2±2.1 (−23.1 to−16.9) | −19.4±2.2 (−23.1 to−15.6) |
| 3-Oxopropanoic, ωC$_3$ | −20.2±2.1 (−23.5 to−17.0) | −24.0±2.5 (−27.7 to − 20.8) | −22.2±3.0 (−27.7 to−17.0) |
| **III. α-Dicarbonyls** | | | |
| Glyoxal, Gly | −16.7±1.7 (−19.4 to−14.0) | −18.1±1.8(−21.3 to −15.2) | −17.4±1.9 (−21.3 to −14.0) |
| Methyglyoxal, mGly | −17.9±1.8 (−21.0 to−15.0) | −19.6±2.0 (−22.5 to−16.5) | −18.8±2.1 (−22.5 to−15.0) |





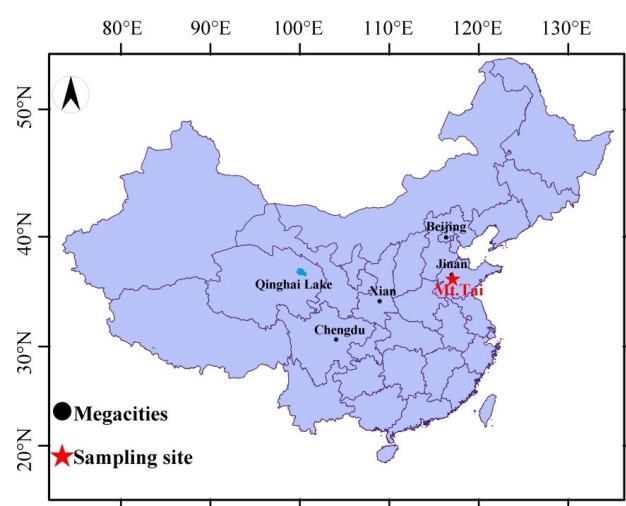

Fig.1. Location of the sampling site (Mt. Tai; 36.25° N, 117.10°E; 1534 m a.s.l.).

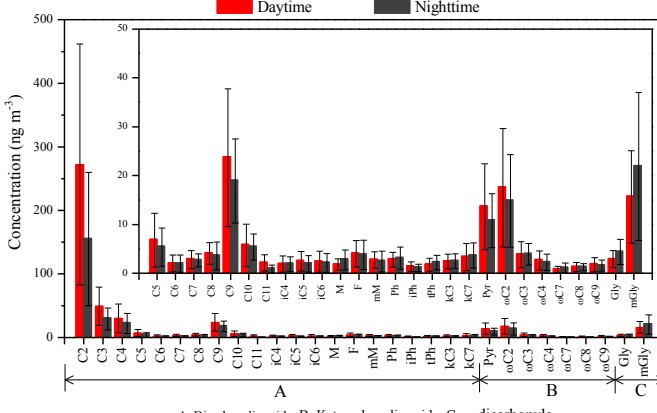


Fig.2. Molecular distributions of dicarboxylic acids and related compounds in PM$_{2.5}$ of Mt.
Tai in North China Plain.



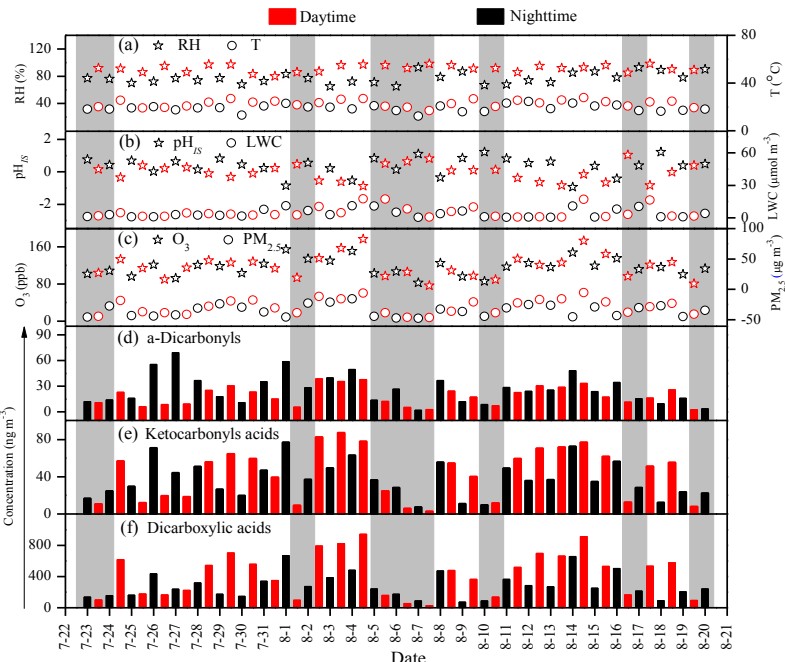


Fig.3. Diurnal variations of relative humidity (RH), temperature (T), in-situ acidity of
particles ($pH_{IS}$), liquid water content of particles (LWC), concentrations of $O_3$, $PM_{2.5}$,
α-dicarbonyls, ketocarboxylic acids, and dicarboxylic acids (rainy days are highlighted in
shadow).





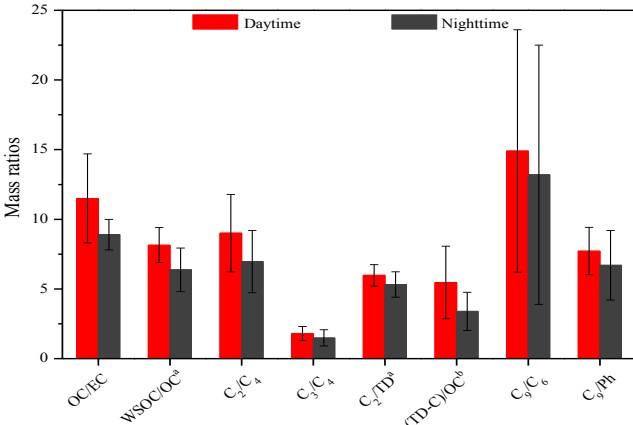


Fig.4. Diurnal variations of mass ratios of OC/EC, WSOC/OC, $C_2/C_4$, $C_3/C_4$, $C_2/TD$,
(TD-C)/OC, $C_9/C_6$, $C_9/Ph$. (TD: total dicarboxylic acids; TD-C: the carbon concentration of
total dicarboxylic acids; [a] the mass ratios expanding 10 times; [b] the mass ratios expanding
100 times).


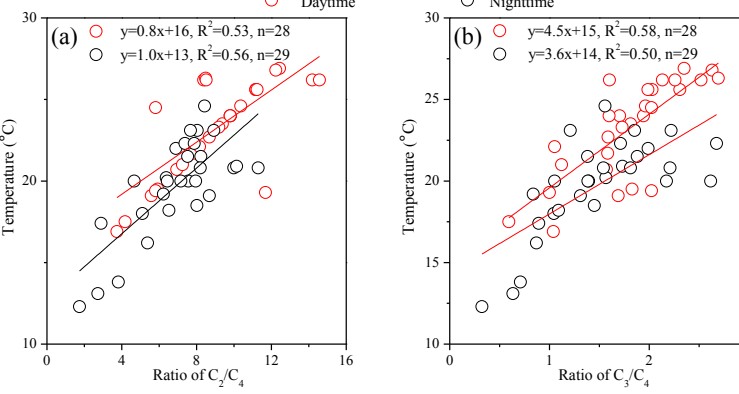


Fig.5. Linear fit regression for temperature (T) with mass ratios of (**a**) $C_2/C_4$ and (**b**) $C_3/C_4$
(See the abbreviations in Table 1).





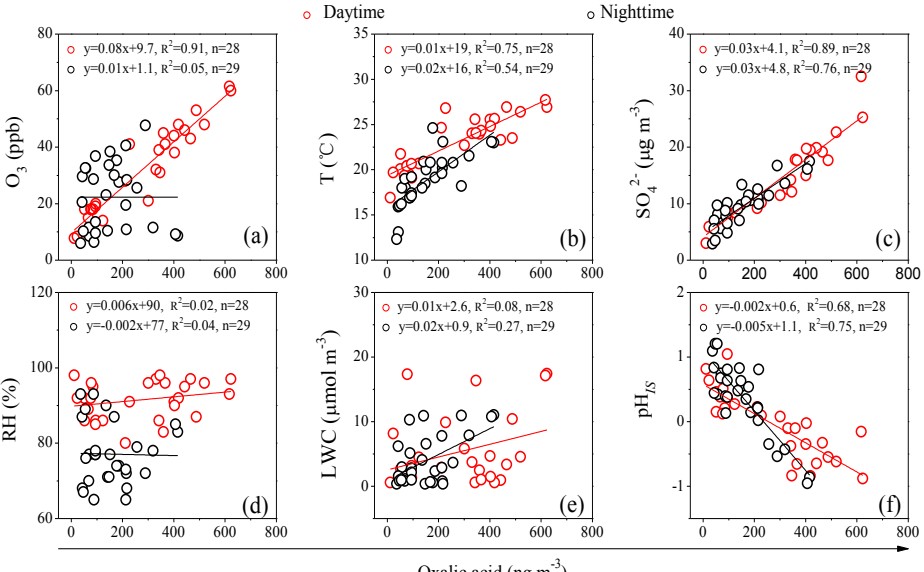

Fig.6. Linear fit regressions of oxalic acid ($C_2$) with (**a**) $O_3$, (**b**) temperature (T), (**c**)$SO_4^{2-}$, (**d**)
relative humidity (RH), (**e**) aerosol liquid water content (LWC), and (**f**) in-situ acidity of
particles ($pH_{IS}$).

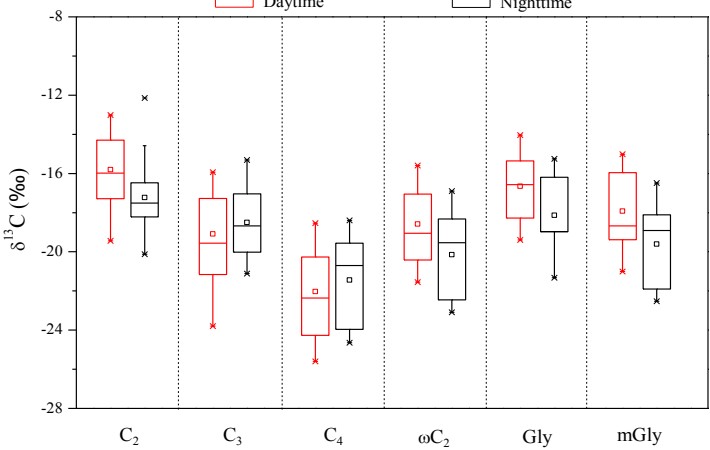

Fig.7. Diurnal variations of stable carbon isotope composition of low molecular weight
dicarboxylic acids ($C_2$−$C_4$), the smallest ketocarboxylic acids ($\omega C_2$) and α-dicarbonyls (Gly,
mGly) in $PM_{2.5}$ collected at the summit of Mt. Tai during the summer.



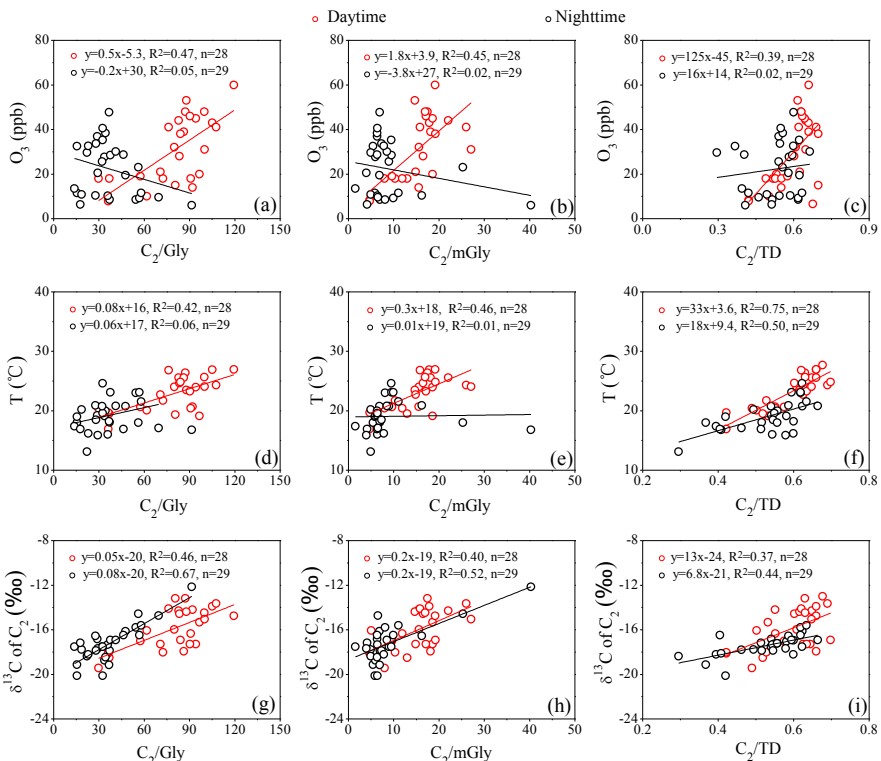


Fig.8. Correlation analysis for the mass ratios of $C_2$/Gly, $C_2$/mGly and $C_2$/TD with (**a-c**)
concentrations of $O_3$; (**d-f**) temperature and (**g-i**) $\delta^{13}C$ of $C_2$ during the daytime and
nighttime ($C_2$/TD: mass ratio of oxalic acid to total dicarboxylic acids; T: temperature).



