# Peer review of "Molecular distribution and stable carbon isotopic compositions 1 of dicarboxylic acids and related SOA from biogenic sources in 2 the summertime atmosphere of Mt. Tai in the North China Plain 3 4 5 Jingjing Meng1,3, Gehui Wang2,3,4\*, Zhanfang Hou1,3, Xiaodi Liu<su"

_Atmospheric Chemistry and Physics, 2018_

## Referee Comment (RC1) · Anonymous Referee #3 · 14 Jul 2018

Review of "Molecular distribution and stable carbon isotopic compositions of dicarboxylic acids and related SOA from biogenic sources in the summertime atmosphere of Mt. Tai in the North China Plain" by Meng et al.

This manuscript reported the concentrations and isotopic compositions of dicarboxylic acids in the summertime of Mt. Tai. The bulk of the results presented focuses on the following aspects: (1) comparing the concentrations of species and the ratios of different dicarboxylic acids between day and night; (2) correlating the species concentrations with other parameters (T, $O_3$, RH, pH, etc); (3) the $\delta^{13}C$ of the dicarboxylic acids. The authors concluded that the dicarboxylic acids in Mt. Tai, a forested area with high elevation, mainly originate from local photochemical oxidation of biogenic emissions. Given the surrounding in Mt. Tai and sampling time, this conclusion is not surprising. Overall, the data analysis is routine and the conclusion is generally supported by experimental evidence. I recommend publication after major revisions.

Major Comments

1.      My major concern is the lack of innovation of this study. As noted in the abstract, a few studies have performed similar measurements and analysis in the same site. The novelty of this manuscript as claimed by the authors is that the measurements were performed in summertime for the first time. It is not well justified that why it is important to conduct the measurements in summer. Also, considering the surroundings of measurement site and sampling time (i.e., summer), isn't the conclusion largely expected? It is not clear what are the scientific questions that authors aimed to address.

2.      Another major comment is regarding the comparison between daytime and nighttime. Due to the high elevation, the sampling site is in the residual layer at night. At night, the residual layer is isolated from surface emissions. The dicarboxylic acids and other species at night are mainly carried over from late afternoon formation. Also, the late afternoon mixing ratios of BVOCs, NOx and O3 govern overnight chemistry within the residual layer. Please refer to Edwards et al. (2017). Assuming no further oxidation at night, the species concentrations are expected to be similar to that in the late afternoon. Thus, the day (8am-8pm) vs night (8pm-night day 8am) comparison may

merely reflect the difference in species concentration between noon and 6pm (when sampling site is above the boundary layer).

This study is an excellent opportunity to study the residual layer chemistry. However, in current most of discussions (Line 184, Line 223, Line 298, etc), the format is like "something is higher in daytime than nighttime, suggesting stronger photochemistry or stronger biogenic activity in the day". The authors should clearly state the physical model behind the day vs. night comparison and better explain the observations (including the $NH_4NO_3$ concentration in Line 184). The authors are encouraged to look in depth into the day vs. night comparison.

3.      Line 193-194. Why is LWC higher in daytime than nighttime?

4.      Line 243-248. The discussions on the higher concentration of dicarbonyls at night is confusing. Firstly, dicarbonyls are produced from photooxidation of isoprene and other VOCs, not aqueous phase reaction. Secondly, how does the "opposite pattern" suggest "aerosol aqueous phase oxidation"? Thirdly, I believe "impressed" is a typo. In light of my previous comments, the authors should explore more reasons in terms of this day vs night comparison.

5.      Line 268-269. Please provide evidence that C2/C4 and C3/C4 can be regarded as indicators of photochemical aging. Also, the measured C2/C3 and C2/C4 ratios are actually very close to that from vehicle exhausts. This is contradictory to the conclusion that dicarboxylic acids are mainly from biogenic emissions.

6.      Line 279-282. This is one of many cases that the authors need to better explain the link between evidence and conclusion. Why would the "correlation between C2/C4 and temperature at night" suggest "local photochemical oxidation"?

7.      Line 360-362. The role of nighttime chemistry is not justified at all. The lack of correlation between BSOA tracers and $O_3$ at night could be attributed to that most of BSOA tracers at night are carried over from later afternoon product and the $O_3$ at night may be quickly titrated by residual NO. Again, this comment falls in the scope of day vs. night comparison.

Minor Comments

1.      Line 61. "water" is uncountable noun. Replace "waters" with "water".

2.      Line 132. What type is the GC column?

3.      Line 304. The change in topic is too quick. Rephrase the sentence to improve the transition.

4.      Line 360. As mentioned above in the text, it is not only from $O_3$ oxidation, also OH oxidation.

5.      Line 377. Replace "linger" with "linear".

Reference

Edwards, P. M., Aikin, K. C., Dube, W. P., Fry, J. L., Gilman, J. B., de Gouw, J. A., Graus, M. G., Hanisco, T. F., Holloway, J., Hübler, G., Kaiser, J., Keutsch, F. N., Lerner, B. M., Neuman, J. A., Parrish, D. D., Peischl, J., Pollack, I. B., Ravishankara, A. R., Roberts, J. M., Ryerson, T. B., Trainer, M., Veres, P. R., Wolfe, G. M., Warneke, C., and Brown, S. S.: Transition from high- to low-NOx control of night-time oxidation in the southeastern US, Nat Geosci, 10, 490, 10.1038/ngeo2976, 2017.

---

## Referee Comment (RC2) · Anonymous Referee #4 · 21 Jul 2018

This manuscript summarizes results from a summertime study at mt. Tai in China where aerosol samples, ozone, and environmental parameters were measured. The authors determine daytime/nighttime concentrations and $\delta^{13}C$ of various carboxylic acids in an effort to characterize the role of bVOCs on SOA formation and aging. The paper is overall well-written although some parts need improvements. My major concern is about interpretation of the daytime/nighttime data. First of all, the standard deviations of the average values presented throughout the paper are rather large which mean that although the average values may be slightly different during day and night, statistically there's not a significant difference between the observations. These need to be addressed/corrected throughout the paper. Secondly, as indicated in L171-172, nighttime data represent free tropospheric measurements, meaning whatever was left in the residual layer from daytime, so nighttime observations aren't truly representing nighttime emissions/oxidations. The most unique aspect of the paper is the $\delta^{13}C$ analysis and interpretation of the results. I therefore support publishing the manuscript with major revisions after the authors have addressed my comments above as well as the other scientific comments and editorial suggestions listed below.

**Scientific Comments:**

- L40: define C6, Ph, Gly, and mGly (also in L249). Also how are the ratios mentioned here expected to behave for anthropogenic precursors?
- L41: how is 'related SOA with biogenic precursor' defined/determined?
- L164: What's the influence of organic acids on pH? Since the AIM model incorporates only the major inorganic ions, how do you think your pH results are affected by the presence of organic acids?
- L177: Looking at the observed variability in the values presented in Table 1, there isn't a significant change between daytime and nighttime concentrations although the average for some species is higher during daytime. I think this point needs to be clarified here and throughout the paper.
- L217: is the vegetation (tree types) also similar at this location and Mt. Fuji? I think that's more important rather than latitude and altitude of the sites.
- L243-245: Based on the average values in the table, total a-dicarbonyls were actually higher in daytime, so I don't think the data support the statement mentioned here. Do you mean only glyoxal and methyl glyoxal? Also based on the graph, it seems on most days the difference between daytime and nighttime total conc. of a-dicarbonyls was minor, so the pattern you're highlighting is not consistent. Please clarify.
- L265-267: I'm not understanding the difference between the beginning statement and the second part of the sentence. C4 is still a longer-chain diacid compared to C3. Please clarify and be more precise in what constitutes a longer-chain diacid.
- L269-275: Again it seems that given the variability of the observed diacid ratios in this study and those of previous studies, there's no significant difference between observations at different photochemical ages. I'm not convinced the conclusions regarding age are strongly supported by the data.
- L275-277: Photochemical oxidation is stronger compared to what? Nighttime or aqueous oxidation? Please clarify.
- L298-299: again given the variability observed in the daytime/nighttime data, the difference in the average values isn't significant.

- L328: what are the SOA tracers from these compounds? It will be useful to add to the legend in Table 4 what parent hydrocarbon is associated with each tracer.
- L339: some of the acids related to apinene and bcaryophyllene oxidation also correlate well with the diacids, so why only highlight isoprene contribution to SOA? In fact, the following sentence indicates that bSOA in general control production of the diacids, so perhaps it's better to combine these few sentences together.
- L360: I disagree with the statement that daytime ozone oxidation of isoprene and apinene was more important than OH oxidation of these compounds. Lifetime of these bVOCs even at background OH levels is a lot shorter than with respect to ozone oxidation. The observed correlation is just a correlation and not a causality. Related to this, I think the conclusion in the sentence starting in L454-455 needs to be removed.
- L366-368: SOA formation potential under different oxidants are also different, and so can contribute to the higher observed tracers during daytime.
- L381-382: In addition to the similar formation pathways (aqueous oxidation) for sulfate and oxalic acid, since oxalic acid formation is acid-catalyzed, one expects to have a good correlation with sulfate and oxalic acid (since the site is far from agricultural sources, I'm assuming most of the sulfate is acidic). Please add this discussion as a contributor to the good correlation as well.
- L393: aerosol composition is also very important for determining LWC of aerosols at a given RH.
- L428-429: Please indicate here specifically what trends in the ratios would suggest aging.
- L440-442: I'm a bit confused about this sentence. Higher values of glyoxal and methyl glyoxal relative to what? Please clarify. Also, from sentence above, I was under the impression that freshly emitted BVOCs are depleted in 13C, so why do the authors indicate that Gly and mGly are formed from oxidation of bVOCs enriched in 13C?
- L448 (also in the abstract): Indicate that 'average' concentration of some species are higher in the day compared to night since as mentioned above, the variability in the measured data was too high to conclude beyond the average.

**Minor/Editorial Comments:**

- L56: consider changing to "…, of which up to 80% are water soluble".
- L73: "… C2 is largely produced…"
- L80: change "liquid water content-enriched aerosol" to "aqueous aerosol"
- L87: change "independent" to "isolated"
- L 88: change "One of the severest air-polluted regions" to "one of the regions with worst air pollution in the world"
- L90: change "few information" to "little information"
- L109: indicate also the altitude of the sampling site in the main text
- L121: "site"
- L152 ad L163: "in-situ particle PH"
- L159: add "… to remove insoluble particles…"
- L222: replace "difference" with "pattern"
- L238: The sentence is too long. Consider starting a new sentence after the references.
- L246: consider replacing "precursors" to "compounds"
- L248: consider replacing "impressed" with "more significant"

- L258: "by wet deposition"
- L265: "by photochemical degradation"
- L278: delete 'would'
- L282: "at Mt. Tai"
- L289: consider changing "troposphere" to "atmosphere"
- L377: "linear"
- L447: either "ground" or "surface"; probably don't need to have both words
- Table1: is the upper end of RH at night 93% rather than 193%?
- Change the order of Fig. 6 and Fig. 5 as Fig. 6 is referred to before Fig. 5. Also it seems the next Figure that authors refer to is Fig. 8. Please use the figures in the same order they appear in the text.

---

## Author Comment (AC1) · 31 Aug 2018

Anonymous Referee #3 General comments: 1. Comments: This manuscript reported the concentrations and isotopic compositions of dicarboxylic acids in the summertime of Mt. Tai. The bulk of the results presented focuses on the following aspects: (1) comparing the concentrations of species and the ratios of different dicarboxylic acids between day and night; (2) correlating the species concentrations with other parameters (T, O3, RH, pH, etc); (3) the $\delta$13C of the dicarboxylic acids. The authors concluded that the dicarboxylic acids in Mt. Tai, a forested area with high elevation, mainly originate from local photochemical oxidation of biogenic emissions. Given the surrounding in Mt. Tai and sampling time, this conclusion is not surprising. Overall, the data analysis is routine and the conclusion is generally supported by experimental evidence. I recommend publication after major revisions.

Response: We thank the reviewer for the helpful comments above and below.

Major Comments: 2. Comments: My major concern is the lack of innovation of this study. As noted in the abstract, a few studies have performed similar measurements and analysis in the same site. The novelty of this manuscript as claimed by the authors is that the measurements were performed in summertime for the first time. It is not well justified that why it is important to conduct the measurements in summer. Also, considering the surroundings of measurement site and sampling time (i.e., summer), isn't the conclusion largely expected? It is not clear what are the scientific questions that authors aimed to address.

Response: (1) The innovation of this study is described as follows: Firstly, several field measurements had been conducted to investigate the molecular compositions, sources and formation mechanisms of SOA including dicarboxylic acids at Mt. Tai, but most of them were performed in May and June and mainly focused on the impact of anthropogenic activities such as field burning of wheat straw, while no information on dicarboxylic acids and related SOA in Mt. Tai during the typical summertime season (i.e., July and August) is available when the emission of biogenic volatile organic compounds (BVOCs) is dominant. A 3-D model simulation shows that about 79% of oxalic acid in the global atmosphere is originated from the oxidation of natural vegetation emissions (Myriokefalitakis et al., 2011), suggesting the dominant contribution of BVOCs to the global SOA loading. Therefore, it is necessary to investigate the abundances, compositions and formation mechanisms of oxalic acid and related SOA when vegetation emission is dominant, especially in the forested highland region where aerosols are more accessible to clouds due to higher elevation (See Page 4, Line 91-105). Secondly, compound-specific stable carbon isotope analysis is a powerful tool

to provide important information of the sources and atmospheric processing of organic aerosols. Analyses of stable carbon isotope ratios of dicarboxylic acids and related SOA can be effectively applied to assessing the photochemical aging level and relative contributions of primary emissions to aerosols in the atmosphere. To our best knowledge, characteristics of the stable carbon isotopic compositions of dicarboxylic acids and related SOA in mountainous regions have not been investigated before. The current work is for the first time to report the stable carbon isotopic compositions of dicarboxylic acids and related compounds in a mountainous area, which are very helpful for improving our understanding on the sources, formation mechanisms and atmospheric behavior of SOA (Please see Page 4-5, Line 106-118). (2) The scientific questions that we aimed to address were listed as follows: The scientific questions that we aimed to address are sources and formation mechanisms of oxalic acid and related SOA in the free troposphere over Mt. Tai (Please see Page 5, Line 118-123).

3. Comments: Another major comment is regarding the comparison between daytime and nighttime. Due to the high elevation, the sampling site is in the residual layer at night. At night, the residual layer is isolated from surface emissions. The dicarboxylic acids and other species at night are mainly carried over from late afternoon formation. Also, the late afternoon mixing ratios of BVOCs, NOx and O3 govern overnight chemistry within the residual layer. Please refer to Edwards et al. (2017). Assuming no further oxidation at night, the species concentrations are expected to be similar to that in the late afternoon. Thus, the day (8am-8pm) vs night (8pm-night day 8am) comparison may merely reflect the difference in species concentration between noon and 6pm (when sampling site is above the boundary layer). This study is an excellent opportunity to study the residual layer chemistry. However, in current most of discussions (Line 184, Line 223, Line 298, etc), the format is like "something is higher in daytime than nighttime, suggesting stronger photochemistry or stronger biogenic activity in the day". The authors should clearly state the physical model behind the day vs. night comparison and better explain the observations (including the NH4NO3 concentration in Line 184). The authors are encouraged to look in depth into the day vs. night comparison.

Response: We agree the comments above that the current work is an excellent opportunity to study the residual layer chemistry. Oxalic acid in the mountain atmosphere showed a strong linear correlation with temperature (Figure 6b) during the day and night, suggesting a dependent relationship of oxalic acid production with temperature. As the temperature was constantly variable and O3 level (22±12 ppb, Table 1) was high throughout the night at Mt. Tai, thus we assume that both gas-phase and aerosol-phase oxidations should continuously proceed throughout the whole night. As suggested by the reviewer, we did more analysis to discuss the potential factors causing the day and night differences in sulfate, nitrate and ammonium (see Figure S1 and Page 8-9, Line 223-233). Sulfate is mainly produced from aqueous phase oxidation of SO2, which is favored by higher temperature and humid conditions. Thus, concentration and relative abundance of sulfate are higher in daytime than in nighttime (Table 1 and Fig. S1). Particulate nitrate is mainly formed via gas phase oxidation of NO2 with OH radical and subsequent partitioning into aerosol phase with ammonia to form NH4NO3. NH4NO3 is volatile and thus lower temperature at night is favorable for NH4NO3 enriching in aerosol phase, resulting in NH4NO3 more abundant at night during the Mt. Tai observation period. These discussions were added into the revised manuscript.

4. Comments: Line 193-194. Why is LWC higher in daytime than nighttime?

Response: Aerosol LWC is controlled by the ambient relative humidity (Clegg et al., 1998) and the concentrations of inorganic salt (Fountoukis and Nenes, 2007). As shown in Table 1, the total concentration (21.7ïĆś11.5 $\mu$g m-3) of SO42-, NO3- and NH4+ during the daytime was almost equivalent to that (20.4ïĆś8.2 $\mu$g m-3) during the nighttime, but the relative humidity (92±5.0%) in daytime was higher than that in nighttime (77±8.2%). Therefore, the concentration of LWC in daytime was higher than that in nighttime. We have added these discussions into the revised manuscript. Please see Page 9, Line 239-245.

5. Comments: Line 243-248. The discussions on the higher concentration of dicarbonyls at night is confusing. Firstly, dicarbonyls are produced from photooxidation of

isoprene and other VOCs, not aqueous phase reaction. Secondly, how does the "opposite pattern" suggest "aerosol aqueous phase oxidation"? Thirdly, I believe "impressed" is a typo. In light of my previous comments, the authors should explore more reasons in terms of this day vs night comparison.

Response: Suggestion taken. Dicarbonyls in the aerosol phase are produced by the gas-phase photooxidation of isoprene and other VOCs and the subsequent partitioning into the aerosol aqueous phase. The higher concentrations of dicarbonyls at night can in part be attributed to the nighttime lower temperatures, which are favorable for the partitioning of gaseous glyoxal and methylglyoxal into the aerosol phase. We are sorry that the "impressed" word here is a typo. We have revised the statements, please see Page 11-12, Line 302-310.

6. Comments: Line 268-269. Please provide evidence that C2/C4 and C3/C4 can be regarded as indicators of photochemical aging. Also, the measured C2/C3 and C2/C4 ratios are actually very close to that from vehicle exhausts. This is contradictory to the conclusion that dicarboxylic acids are mainly from biogenic emissions.

Response: We are sorry for the mistake about the mass ratios of C2/C4 and C3/C4 from vehicle exhausts. Kawamura and Kaplan (1987) have reported that the ratios of C2/C4 and C3/C4 from vehicle exhausts were 4.1 and 0.35, respectively (see Page 12, Line 334- 335). The C2/C4 and C3/C4 ratios in this study are $8.0 \pm 2.7$ and $1.6 \pm 0.6$, respectively, higher than those in aerosols freshly emitted from vehicle exhausts. In addition, elemental carbon (EC) concentrations were very low and similar between day and night during the observation period (Table 1), suggesting that the impact of pollutants derived from anthropogenic sources including vehicle exhausts during the campaign was negligible. Therefore, traffic emission is not expected to have a large contribution to dicarboxylic acids in this study. It's consistent with the conclusion that dicarboxylic acids in Mt. Tai during summer are mainly from biogenic emissions rather than anthropogenic emissions.

7. Comments: Line 279-282. This is one of many cases that the authors need to better explain the link between evidence and conclusion. Why would the "correlation between C2/C4 and temperature at night" suggest "local photochemical oxidation"?

Response: Decomposition of C4 and/or C3 into C2 is one of the major formation pathways of oxalic acid, which is favored by temperature. Thus, a linear correlation of C2/C4 or C2/C3 with temperature has been frequently observed (Kawamura and Ikushima, 1993; Meng et al., 2013; Pavuluri et al., 2010). Temperature measured at the sampling site is a meteorological parameter, which only reflects the local meteorological conditions rather than the upwind conditions. Therefore, a significant correlation between C2/C4 and temperature can only be observed when SOA is largely derived from local precursor oxidation rather than from long-range transport. We have added related discussions into the text, please see Page 13, Line 341-350.

8. Comments: Line 360-362. The role of nighttime chemistry is not justified at all. The lack of correlation between BSOA tracers and O3 at night could be attributed to that most of BSOA tracers at night are carried over from later afternoon product and the O3 at night may be quickly titrated by residual NO. Again, this comment falls in the scope of day vs. night comparison.

Response: Thanks for your suggestion. We have revised the related contents in the revised manuscript. Please see Page 16, Line 432-441.

Minor Comments:

9. Comments: Line 61. "water" is uncountable noun. Replace "waters" with "water".

Response: Suggestion taken. Please see Page 3, Line 63.

10. Comments: Line 132. What type is the GC column?

Response: The type of the GC column is fused silica capillary column (HP-5, 0.2 mm $\times$ 25 m, film thickness 0.5 $\mu$m). We have added the details about the type of the GC column into the revised manuscript. Please see Lage 6, Line 153-156.

11. Comments: Line 304. The change in topic is too quick. Rephrase the sentence to improve the transition.

Response: Suggestion taken. We have rephrased the sentence as follows: However, the average values of C9/C6 (14±9.0) and C9/Ph (7.2±2.2) at the mountaintop of Mt. Tai are higher than those in urban regions such as Xi'an, China (C9/C6: 3.1; C9/Ph: 5.6) (Cheng et al., 2013) and also higher than those in other mountainous during summer such as Mt. Himalayan, India (C9/C6: 2.1; C9/Ph: 0.2) (Hedge and Kawamura, 2012) and Mt. Fuji, Japan (C9/C6: 3.1) (Mochizuki et al., 2017), indicating the important contribution of biogenic sources to SOA in the Mt. Tai region. Model simulation (Fu et al., 2008) and field observations (Meng et al., 2014) have suggested that. . . . . .. Please see Page 14, Line 371-377.

12. Comments: Line 360. As mentioned above in the text, it is not only from O3 oxidation, also OH oxidation.

Response: Thanks for your suggestion. We have added "and OH radicals" in the old sentences. Please see Page 16, Line 436.

13. Comments: Line 377. Replace "linger" with "linear".

Response: Suggestion taken. Please see Page 17, Line 463.

References:

Carlton, A. G., Turpin, B. J., Lim, H.-J., Altieri, K. E., and Seitzinger, S.: Link between isoprene and secondary organic aerosol (SOA): Pyruvic acid oxidation yields low volatility organic acids in clouds, Geophysical Research Letters, 33, L06822, 10.1029/2005gl025374, 2006.

Carlton, A. G., Turpin, B. J., Altieri, K. E., Seitzinger, S., Reff, A., Lim, H.-J., and Ervens, B.: Atmospheric oxalic acid and SOA production from glyoxal: Results of aqueous photooxidation experiments, Atmospheric Environment, 41, 7588-7602, http://dx.doi.org/10.1016/j.atmosenv.2007.05.035, 2007.

[Figure]

Carlton, A. G., Wiedinmyer, C., and Kroll, J. H.: A review of Secondary Organic Aerosol (SOA) formation from isoprene, Atmos. Chem. Phys., 9, 4987-5005, 10.5194/acp-9-4987-2009, 2009.

Cheng, C., Wang, G., Zhou, B., Meng, J., Li, J., Cao, J., and Xiao, S.: Comparison of dicarboxylic acids and related compounds in aerosol samples collected in Xi'an, China during haze and clean periods, Atmospheric Environment, 81, 443-449, http://dx.doi.org/10.1016/j.atmosenv.2013.09.013, 2013. Clegg, S. L., Brimblecombe, P., and Wexler, A. S.: Thermodynamic Model of the System H+$-$NH4+$-$SO42-$-$NO3- $-$H2O at Tropospheric Temperatures, The Journal of Physical Chemistry A, 102, 2137-2154, 10.1021/jp973042r, 1998.

Fountoukis, C., and Nenes, A.: ISORROPIA II: a computationally efficient thermodynamic equilibrium model for K+$-$Ca2+$-$Mg2+$-$NH4+$-$Na+$-$SO24$-$$-$NO3$-$$-$Cl$-$$-$H2O aerosols, Atmos. Chem. Phys., 7, 4639-4659, 10.5194/acp-7-4639-2007, 2007.

Fu, T.-M., Jacob, D. J., Wittrock, F., Burrows, J. P., Vrekoussis, M., and Henze, D. K.: Global budgets of atmospheric glyoxal and methylglyoxal, and implications for formation of secondary organic aerosols, J. Geophys. Res., 113, D15303, 10.1029/2007jd009505, 2008.

Kawamura, K., and Ikushima, K.: Seasonal changes in the distribution of dicarboxylic acids in the urban atmosphere, Environmental Science & Technology, 27, 2227-2235, 10.1021/es00047a033, 1993.

Kawamura, K., Okuzawa, K., Aggarwal, S. G., Irie, H., Kanaya, Y., and Wang, Z.: Determination of gaseous and particulate carbonyls (glycolaldehyde, hydroxyacetone, glyoxal, methylglyoxal, nonanal and decanal) in the atmosphere at Mt. Tai, Atmos. Chem. Phys., 13, 5369-5380, 10.5194/acp-13-5369-2013, 2013a.

Kawamura, K., Tachibana, E., Okuzawa, K., Aggarwal, S. G., Kanaya, Y., and Wang, Z. F.: High abundances of water-soluble dicarboxylic acids, ketocarboxylic acids and $\alpha$-

dicarbonyls in the mountain aerosols over the North China Plain during wheat burning season, Atmos. Chem. Phys. , 13, 3695-3734, 10.5194/acpd-13-3695-2013, 2013b.

Li, J. J., Wang, G. H., Cao, J. J., Wang, X. M., and Zhang, R. J.: Observation of biogenic secondary organic aerosols in the atmosphere of a mountain site in central China: temperature and relative humidity effects, Atmos. Chem. Phys., 13, 11535-11549, 10.5194/acp-13-11535-2013, 2013.

Meng, J., Wang, G., Li, J., Cheng, C., Ren, Y., Huang, Y., Cheng, Y., Cao, J., and Zhang, T.: Seasonal characteristics of oxalic acid and related SOA in the free troposphere of Mt. Hua, central China: Implications for sources and formation mechanisms, Science of The Total Environment, 493, 1088-1097, http://dx.doi.org/10.1016/j.scitotenv.2014.04.086, 2014.

Pavuluri, C. M., Kawamura, K., and Swaminathan, T.: Water-soluble organic carbon, dicarboxylic acids, ketoacids, and $\alpha$-dicarbonyls in the tropical Indian aerosols, Journal of Geophysical Research: Atmospheres, 115, D11302, 10.1029/2009jd012661, 2010.

Zhu, Y., Yang, L., Chen, J., Kawamura, K., Sato, M., Tilgner, A., van Pinxteren, D., Chen, Y., Xue, L., Wang, X., Herrmann, H., and Wang, W.: Molecular distributions of dicarboxylic acids, oxocarboxylic acids, and $\alpha$-dicarbonyls in PM2.5 collected at Mt. Tai, in North China in 2014, Atmos. Chem. Phys. Discuss., 2018, 1-31, 10.5194/acp-2017-1240, 2018.

---

## Author Comment (AC2) · 31 Aug 2018

General comments:

1. Comments: This manuscript summarizes results from a summertime study at Mt. Tai in China where aerosol samples, ozone, and environmental parameters were measured. The authors determine daytime/nighttime concentrations and ïĄd'13C of various carboxylic acids in an effort to characterize the role of bVOCs on SOA formation and

aging. The paper is overall well-written although some parts need improvements. My major concern is about interpretation of the daytime/nighttime data. First of all, the standard deviations of the average values presented throughout the paper are rather large which mean that although the average values may be slightly different during day and night, statistically there's not a significant difference between the observations. These need to be addressed/corrected throughout the paper. Secondly, as indicated in L171-172, nighttime data represent free tropospheric measurements, meaning whatever was left in the residual layer from daytime, so nighttime observations aren't truly representing nighttime emissions/oxidations. The most unique aspect of the paper is the ïA̧d'13C analysis and interpretation of the results. I therefore support publishing the manuscript with major revisions after the authors have addressed my comments above as well as the other scientific comments and editorial suggestions listed below.

Response: We thank the reviewer for the helpful comments above. In the revised version, we have performed a statistic test (i.e., Student's t-test) to verify if the day and night aerosol chemistry is of significant difference. As shown in the Table S1, the concentrations and compositions of major species in PM2.5 between day and night show a P value less than 0.005, which clearly demonstrates that the abundances and compositions of the major species during the day and night are statistically different. Figure S1 also shows that during the nighttime sulfate decreased while nitrate and ammonium increases. Such a diurnal change in inorganic ion compositions further suggests the significant difference in aerosol chemistry between day and night. Related statements have been added into the text. Please see Line 211-217 in Page 8 and Line 223-233 in Page 8-9, respectively.

Scientific Comments:

2. Comments: L40: define C6, Ph, Gly, and mGly (also in L249). Also how are the ratios mentioned here expected to behave for anthropogenic precursors?

Response: Suggestion taken. We have defined C6, Ph, Gly, and mGly in the revised

manuscript. Please see Line 42-43 in Page 2 and Line 310-311 in Page 12, respectively. C6 and Ph are believed to be formed via secondary oxidations of anthropogenic cyclic olefins (e.g., cyclohexene) (Hatakeyama et al., 1987) and aromatic hydrocarbons, respectively. C9 is mainly produced from photochemical oxidation of oleic acid, which is a biogenic unsaturated fatty acid containing a double bond at the C-9 position (Wang et al., 2010). Therefore, both ratios of C9/C6 and C9/Ph are indicative of the source strengths of biogenic versus anthropogenic emissions. Please see Line 362–370 in Page 13-14 of the revised manuscript. Model simulation (Fu et al., 2008) and field observations (Meng et al., 2014) have suggested that the concentration ratio of particulate Gly/mGly is about 1:5 when biogenic sources are predominant and is about 1:1 when anthropogenic sources are predominant. Therefore, the mass ratio of particulate Gly/mGly is also indicative of the source strengths of biogenic versus anthropogenic emissions. Please see Page 14, Line 377-380.

3. Comments: L41: how is 'related SOA with biogenic precursor' defined/ determined?

Response: Thanks for your suggestion. "related SOA" is defined as "major dicarboxylic acids and related SOA", which consists of major dicarboxylic acids, ketocarboxylic acids and $\alpha$-dicarbonyls (i.e., C2, C3, C4, $\omega$C2, Pyr, Gly and mGly), while "biogenic precursor" is defined as "SOA tracers derived from isoprene-, $\alpha$-/$\beta$-pinene- and $\beta$-caryophyllene". We have added these definitions into the revised manuscript. Please see Page 2, Line 42-46. As for the analysis method of biogenic precursors, we have added the related information as follows: The analysis method of biogenic precursors has been reported elsewhere (Li et al., 2013). Briefly, one fourth of the filter was cut and extracted with a mixture of dichloromethane and methanol (2:1, v/v) under ultrasonication. The extracts were concentrated using a rotary evaporator under vacuum conditions and then blow down to dryness using pure nitrogen. After reaction with a mixture of N,O-bis-(trimethylsilyl) trifluoroacetamide (BSTFA) and pyridine (5:1, v/v) at 70°C for 3 h. Biogenic secondary organic aerosol (BSOA) tracers in the derivatized samples were determined by GC-MS. These data were used in this study to explore

the biogenic sources of dicarboxylic acids and related SOA. Please see Page 6-7, Line 165-173.

4. Comments: L164: What's the influence of organic acids on pH? Since the AIM model incorporates only the major inorganic ions, how do you think your pH results are affected by the presence of organic acids?

Response: The Extended AIM Thermodynamic Model (E-AIM, Model II, http:// www.aim. env.uea.ac.uk/aim/) was employed to calculate aerosol liquid water content (LWC) and in-situ particle pH (pHIS). E-AIM II is an equilibrium thermodynamic model that can simulate liquid and solid phase of ionic compositions accurately in the $SO_4^{2-}$– $NO_3^-$ $NH_4^+$-$H^+$ system under certain temperature and relative humidity (Clegg et al., 1998a; Li et al., 2013). Compared with $SO_4^{2-}$, $NO_3^-$ and $NH_4^+$, organic acid has little influence on estimation of aerosol acidity due to their low abundance in aerosols (Zhou et al., 2018). Organic acids may contribute free $H^+$ in aerosol aqueous phase and affect partitioning/dissociation of inorganic species in acidic particles. However, Huang et al. (2010) found that oxalic acid, the single most abundant organic acid, contributed little to the free acidity of rain water. This implied that the contribution of organic acids to aerosol free $H^+$ in the Mt. Tai aerosols was most likely minor. Moreover, quantity of free $H^+$ released from organosulfates was proved to be small compared with that estimated from inorganic anion and cation balance. Hygroscopicity of organic compounds are significantly weaker than inorganic species such as $NH_4NO_3$ and $(NH_4)_2SO_4$ (Ansari and Pandis, 2000). Therefore, organic species generally have minor influence on pHIS. So, we did not take into account the organic species, neither did previous studies about the in-situ acidity of aerosol (Li et al., 2013; Meng et al., 2014; Xue et al., 2011). Both field observation and laboratory simulation suggest that oxalic acid is largely derived from the acid-catalyzed heterogeneous oxidation of glyoxal and related precursors in the aqueous phase. As shown in Fig. 6f in the revised manuscript, oxalic acid exhibits a significant negative correlation with pHIS for the daytime ($R^2$=0.68) and nighttime ($R^2$=0.75) samples, respectivelyïijŇsuggesting that acidic conditions are favorable for

the formation of organic acids in the aerosol aqueous phase.

5. Comments: L177: Looking at the observed variability in the values presented in Table 1, there isn't a significant change between daytime and nighttime concentrations although the average for some species is higher during daytime. I think this point needs to be clarified here and throughout the paper.

Response: The variability of the data presented in Table 1 and throughout the paper is large, but our statistic t-test analysis results (Table S1) show that both concentrations and mass ratios between day and night are significantly different. Therefore, our statements, such as "the higher SOA concentrations and mass ratios in daytime are due to the strong photochemical oxidation" and other related discussions, are reasonable. We have also modified the related discussions as follows: OC and WSOC in the PM2.5 samples in daytime are similar to those in nighttime (Table 1), but OC/EC and WSOC/OC ratios are around 1.4 times higher in daytime than in nighttime (Fig. 4), indicating an enhancing SOA production due to the stronger photochemical oxidation in daytime rather than the changes in the boundary layer heights (Hegde and Kawamura, 2012). Please see Page 8, Line 218-222.

6. Comments: L217: is the vegetation (tree types) also similar at this location and Mt. Fuji? I think that's more important rather than latitude and altitude of the sites.

Response: We have checked the vegetation compositions at Mt. Tai and Mt. Fuji, and found both are dominated by broad-leaved forest. We have modified the related statement. Please see Page 10, Line 269-270.

7. Comments: L243-245: Based on the average values in the table, total ïĄą-dicarbonyls were actually higher in daytime, so I don't think the data support the statement mentioned here. Do you mean only glyoxal and methylglyoxal? Also based on the graph, it seems on most days the difference between daytime and nighttime total conc. of ïĄą-dicarbonyls was minor, so the pattern you're highlighting is not consistent. Please clarify.

Response: In the current work, total ïĄą-dicarbonyls means only glyoxal and methyl-glyoxal. As seen in Table 2, concentrations of ïĄą-dicarbonyls were 19 ± 11 ng/m3 in daytime, which were lower than that (27 ± 17 ng/m3) in nighttime. As we mentioned before, we have done a t-test analysis to check if the diurnal difference is of a statistic meaning (Table S1). The t-test results show that concentrations and mass ratios of major species between day and night are of a P-value less than 0.005, which clearly suggests that the diurnal differences are statistically significant.

8. Comments: L265-267: I'm not understanding the difference between the beginning statement and the second part of the sentence. C4 is still a longer-chain diacid compared to C3. Please clarify and be more precise in what constitutes a longer-chain diacid.

Response: Suggestion taken. The longer-chain diacids are defined as the number of carbons contained in the diacids is larger than four such as C5-C11. To clarify this, we have revised the sentence as follows: Previous studies have proposed that the hydroxylation of C4 can be further oxidized into C2 and C3, and C3 can also be oxidized into C2 through intermediate compounds such as hydroxymalonic acid or ketomalonic acid. Please see Page 12, Line 328-330.

9. Comments: L269-275: Again it seems that given the variability of the observed diacid ratios in this study and those of previous studies, there's no significant difference between observations at different photochemical ages. I'm not convinced the conclusions regarding age are strongly supported by the data.

Response: As seen in our previous response, we did a statistic analysis, which shows that the diurnal variation is statistically significant.

10. Comments: L275-277: Photochemical oxidation is stronger compared to what? Nighttime or aqueous oxidation? Please clarify.

Response: Thanks for your suggestion. Photochemical oxidation is stronger in the

daytime than that in the nighttime. We have revised the old sentence as follows: Compared with those in the nighttime, the higher ratios of C2/C4 and C3/C4 (Fig. 4) in the daytime again indicated that the photochemical modification of aerosols is stronger. Please see Page 13, Line 339-341.

11. Comments: L298-299: again given the variability observed in the daytime/nighttime data, the difference in the average values isn't significant.

Response: Here we agree with the comments above. The t-test analysis shows that differences in ratios of C9/C6 and C9/Ph between day and night are not significant. Thus, we have revised the related discussions as follows: As shown in Fig. 4, both ratios of C9/C6 and C9/Ph are similar in the daytime to those in the nighttime. Please see Page 14, Line 370-371.

12. Comments: L328: what are the SOA tracers from these compounds? It will be useful to add to the legend in Table 4 what parent hydrocarbon is associated with each tracer.

Response: Suggestion taken. The SOA tracers from these compounds consist of the secondary organic aerosols (SOA) derived from isoprene, $\alpha$-/$\beta$-pinene and $\beta$-caryophyllene. We have added this explanation to the legend in Table 4. Please see the red words in Page 31, Table 4.

13. Comments: L339: some of the acids related to a-pinene and ïĄć-caryophyllene oxidation also correlate well with the diacids, so why only highlight isoprene contribution to SOA? In fact, the following sentence indicates that bSOA in general control production of the diacids, so perhaps it's better to combine these few sentences together.

Response: Suggestion taken, we have combined these sentences together. Please see Page 15, Line 412-419.

14. Comments: L360: I disagree with the statement that daytime ozone oxidation of isoprene and ïĄą-pinene was more important than OH oxidation of these compounds.

Lifetime of these bVOCs even at background OH levels is a lot shorter than with respect to ozone oxidation. The observed correlation is just a correlation and not a causality. Related to this, I think the conclusion in the sentence starting in L454-455 needs to be removed.

Response: We agree with the reviewer on the comments above. We have revised the discussions about the reason why the lack of correlation was observed between BSOA tracers and O3 at night as follows: These results suggest that the daytime oxalic acid and related SOA in the mountaintop of Mt. Tai are largely derived from O3 and OH radical oxidation of BVOCs such as isoprene and $\alpha$-pinene, while the nighttime oxalic acid and related SOA might be mostly produced by NO3 radical and other oxidizing agents such as H2O2 (Claeys et al., 2004; Herrmann et al., 1999). In addition, the titration of O3 by the residual NO in the nighttime atmosphere could also be responsible for the lack of the correlation between BSOA tracers and O3. Please see Page 16, Line 434-441. We have deleted the conclusion in the sentence starting in L454-455 in the old version manuscript.

15. Comments: L366-368: SOA formation potential under different oxidants are also different, and so can contribute to the higher observed tracers during daytime.

Response: We agree with the comments above that SOA formation potential under different oxidants are different, which could contribute to the higher observed tracers during daytime. As seen in Table 1, O3 concentration during the sampling period is 50% higher in daytime than in nighttime, clearly indicating that oxidation potential in daytime at the Mt. Tai site is stronger. Moreover, isoprene is only emitted by trees during daytime. Thus, we think the higher loadings of BSOA tracers in daytime are caused not only by stronger photochemical oxidation but also by enhanced emissions of BVOCs. We have revised the statement. Please see Page 16-17, Line 446-452.

16. Comments: L381-382: In addition to the similar formation pathways (aqueous oxidation) for sulfate and oxalic acid, since oxalic acid formation is acid-catalyzed, one

expects to have a good correlation with sulfate and oxalic acid (since the site is far from agricultural sources, I'm assuming most of the sulfate is acidic). Please add this discussion as a contributor to the good correlation as well.

Response: Suggestion taken. Please see Page 17, Line See 462-463.

17. Comments: L393: aerosol composition is also very important for determining LWC of aerosols at a given RH.

Response: We agree with the comments above. Aerosol LWC in this study was calculated by using AIM-II model, which considered a $SO_4^{2-}$–$NO_3^-$ $NH_4^+$-$H^+$ system and allowed variable temperature and relative humidity (Clegg et al., 1998b; Li et al., 2013), therefore aerosol LWC is controlled by the ambient relative humidity (Clegg et al., 1998b) and the concentrations of inorganic salt (Fountoukis and Nenes, 2007). As shown in Table 1 of the revised manuscript, the total concentration (21.7ïĆś11.5 $\mu$g m-3) of $SO_4^{2-}$, $NO_3^-$ and $NH_4^+$ during the daytime was almost equivalent to that (20.4ïĆś8.2 $\mu$g m-3) during the nighttime. However, the relative humidity (92$\pm$5.0%) in daytime was higher than that in nighttime (77$\pm$8.2%). Therefore, we only discussed the effect of RH on aerosol LWC. We have changed old sentence as follows: Both RH and aerosol composition are key factors controlling the aerosol LWC. Please see Page 18, Line 480-481.

18. Comments: L428-429: Please indicate here specifically what trends in the ratios would suggest aging.

Response: Suggestion taken. The higher the mass ratios of C2/$\omega$C2, C2/Gly and C2/mGly, the more aged the organic aerosol. To express clearly, we have revised the old sentence as follows: Thus, the higher mass ratios of C2/$\omega$C2, C2/Gly and C2/mGly indicate that organic aerosols are more aged (Wang et al., 2017). Please see Page 19, Line 518-519.

19. Comments: L440-442: I'm a bit confused about this sentence. Higher values of

glyoxal and methyl glyoxal relative to what? Please clarify. Also, from sentence above, I was under the impression that freshly emitted BVOCs are depleted in 13C, so why do the authors indicate that Gly and mGly are formed from oxidation of bVOCs enriched in 13C?

Response: Suggestion taken. The $\delta$13C values of Gly ($-17.4\pm1.9$) and mGly ($-18.8\pm2.1$) were relatively higher than fresh BVOCs such as isoprene (–32‰ $-$ –27‰ emitted directly from vegetation. To express this point clearly, we have revised the related sentence as follows: Therefore, the $\delta$13C values of Gly and mGly are relatively higher than fresh BVOCs such as isoprene, largely attributed to the secondary formation from the oxidation of isoprene and other biogenic precursors. Please see Page 19-20, Line 531-533.

20. Comments: L448 (also in the abstract): Indicate that 'average' concentration of some species are higher in the day compared to night since as mentioned above, the variability in the measured data was too high to conclude beyond the average.

Response: As mentioned above, our t-test analysis showed that the diurnal difference is significant, although the variability in the measured data was very high.

Minor Comments:

21. Comments: L56: consider changing to "..., of which up to 80% are water soluble".

Response: Suggestion taken. Please see Page 3, Line 59.

22. Comments: L73: "... C2 is largely produced...".

Response: Suggestion taken. Please see Page 3, Line 75.

23. Comments: L80: change "liquid water content-enriched aerosol" to "aqueous aerosol".

Response: Suggestion taken. Please see Page 3, Line 81-82.

24. Comments: L87: change "independent" to "isolated".

Response: Suggestion taken. Please see Page 4, Line 89.

25. Comments: L88: change "One of the severest air-polluted regions" to "one of the regions with worst air pollution in the world".

Response: Suggestion taken. Please see Page 4, Line 90.

26. Comments: L90: change "few information" to "little information".

Response: Suggestion taken. Please see Page 4, Line 95-96

27. Comments: L109: indicate also the altitude of the sampling site in the main text.

Response: Thanks for your suggestion. We have added the altitude of the sampling site in the revised manuscript. Please see Page 5, Line 128-130.

28. Comments: L121: "site".

Response: Suggestion taken. We corrected the typo, see Page 4, Line 142.

29. Comments: L152 ad L163: "in-situ particle pH".

Response: Suggestion taken. Please see Page 7, Line 186-187.

30. Comments: L159: add "... to remove insoluble particles...".

Response: Suggestion taken. Please see Page 7, Line 193-194.

31. Comments: L222: replace "difference" with "pattern".

Response: Suggestion taken. Please see Page 10, Line 276.

32. Comments: L238: The sentence is too long. Consider starting a new sentence after the references.

Response: Thanks for your suggestion. We have changed the old sentences as follows: Ketocarboxylic acids are the major intermediates of aqueous phase photochemical oxidation producing dicarboxylic acids in the atmosphere (Kawamura and Ikushima, 1993;Pavuluri and Kawamura, 2016). The concentrations of ketocarboxylic acids are 43 $\pm$ 28 ng m-3 in the daytime and 37 $\pm$ 19 ng m-3 in the nighttime, respectively, with glyoxylic acid ($\omega$C2) being the dominant $\omega$-oxoacid, followed by pyruvic acid (Pyr) and 3-oxobutanoic acid ($\omega$C3) (Table 2 and Fig. 2). Please see Page 11, Line 291-296.

33. Comments: L246: consider replacing "precursors" to "compounds".

Response: Suggestion taken. We have rephrased the sentences, please see Page 11, Line 302-303.

34. Comments: L248: consider replacing "impressed" with "more significant".

Response: Suggestion taken. Please see Page11-12, Line 308-310.

35. Comments: L258: "by wet deposition".

Response: Suggestion taken. Please see Page 12, Line 320-321

36. Comments: L265: "by photochemical degradation".(?)

Response: We have rephrased the expressions as follows: Previous studies have proposed that the hydroxylation of C4 can be further oxidized into C2 and C3, and C3 can also be oxidized into C2 through intermediate compounds such as hydroxymalonic acid or ketomalonic acid Please see Page 12, Line 328-330.

37. Comments: L278: delete 'would'.

Response: Suggestion taken. Please see Page 13, Line 345.

38. Comments: L282: "at Mt. Tai".

Response: Suggestion taken. Please see Page 13, Line 353.

39. Comments: L289: consider changing "troposphere" to "atmosphere".

Response: Suggestion taken. Please see Page 13, Line 361.

40. Comments: L377: "linear".

Response: Suggestion taken. Please see Page 17, Line 463.

41. Comments: L447: either "ground" or "surface"; probably don't need to have both words.

Response: Thanks for your suggestion. We have deleted "surface". Please see Page 20, Line 538-539.

42. Comments: Table1: is the upper end of RH at night 93% rather than 193%?

Response: Sorry for the mistake, we had corrected it. Please see Table 1 in Page 29.

43. Comments: Change the order of Fig. 6 and Fig. 5 as Fig. 6 is referred to before Fig. 5. Also it seems the next Figure that authors refer to is Fig. 8. Please use the figures in the same order they appear in the text.

Response: We disagree on the comments above. Fig. 5 was used to elucidate the dicarboxylic acids and related SOA are mostly derived from the local sources rather than long-range transport, which was discussed in Part 3.3. Fig. 6 was used to investigate the effects of temperature, relative humidity, and O3 concentrations on the formation of oxalic acid and related SOA, which was discussed in Part 3.5. Thus, Fig. 5 should be shown before Fig. 6.

Reference:

Ansari, A. S., and Pandis, S. N.: Water Absorption by Secondary Organic Aerosol and Its Effect on Inorganic Aerosol Behavior, Environmental Science & Technology, 34, 71-77, 10.1021/es990717q, 2000.

Clegg, S. L., Brimblecombe, P., and Wexler, A. S.: Thermodynamic Model of the System H+−NH4+−Na+−SO42-−NO3-−Cl-−H2O at 298.15 K, The Journal of Physical Chemistry A, 102, 2155-2171, 10.1021/jp973043j, 1998a.

[Figure]

Clegg, S. L., Brimblecombe, P., and Wexler, A. S.: Thermodynamic Model of the System $H+-NH4+-SO42--NO3--H2O$ at Tropospheric Temperatures, The Journal of Physical Chemistry A, 102, 2137-2154, 10.1021/jp973042r, 1998.

Fountoukis, C., and Nenes, A.: ISORROPIA II: a computationally efficient thermodynamic equilibrium model for $K+-Ca2+-Mg2+-NH4+-Na+-SO24--NO3--Cl--H2O$ aerosols, Atmos. Chem. Phys., 7, 4639-4659, 10.5194/acp-7-4639-2007, 2007.

Fu, T.-M., Jacob, D. J., Wittrock, F., Burrows, J. P., Vrekoussis, M., and Henze, D. K.: Global budgets of atmospheric glyoxal and methylglyoxal, and implications for formation of secondary organic aerosols, J. Geophys. Res., 113, D15303, 10.1029/2007jd009505, 2008.

Hatakeyama, S., Ohno, M., Weng, J., Takagi, H., and Akimoto, H.: Mechanism for the formation of gaseous and particulate products from ozone-cycloalkene reactions in air, Environmental Science & Technology, 21, 52-57, 10.1021/es00155a005, 1987.

Hegde, P., and Kawamura, K.: Seasonal variations of water-soluble organic carbon, dicarboxylic acids, ketocarboxylic acids, and $\alpha$-dicarbonyls in Central Himalayan aerosols, Atmos. Chem. Phys., 12, 6645-6665, 10.5194/acp-12-6645-2012, 2012.

Kawamura, K., and Ikushima, K.: Seasonal changes in the distribution of dicarboxylic acids in the urban atmosphere, Environmental Science & Technology, 27, 2227-2235, 10.1021/es00047a033, 1993.

Li, J. J., Wang, G. H., Cao, J. J., Wang, X. M., and Zhang, R. J.: Observation of biogenic secondary organic aerosols in the atmosphere of a mountain site in central China: temperature and relative humidity effects, Atmos. Chem. Phys., 13, 11535-11549, 10.5194/acp-13-11535-2013, 2013.

Meng, J., Wang, G., Li, J., Cheng, C., Ren, Y., Huang, Y., Cheng, Y., Cao, J., and Zhang, T.: Seasonal characteristics of oxalic acid and related SOA in the free troposphere of Mt. Hua, central China: Implications for sources

and formation mechanisms, Science of The Total Environment, 493, 1088-1097, http://dx.doi.org/10.1016/j.scitotenv.2014.04.086, 2014.

Pavuluri, C. M., and Kawamura, K.: Enrichment of (13)C in diacids and related compounds during photochemical processing of aqueous aerosols: New proxy for organic aerosols aging, Scientific Reports, 6, 36467, 10.1038/srep36467, 2016.

Wang, G., Xie, M., Hu, S., Gao, S., Tachibana, E., and Kawamura, K.: Dicarboxylic acids, metals and isotopic compositions of C and N in atmospheric aerosols from inland China: implications for dust and coal burning emission and secondary aerosol formation, Atmos. Chem. Phys., 10, 6087-6096, 10.5194/acp-10-6087-2010, 2010.

Wang, J., Wang, G., Gao, J., Wang, H., Ren, Y., Li, J., Zhou, B., Wu, C., Zhang, L., Wang, S., and Chai, F.: Concentrations and stable carbon isotope compositions of oxalic acid and related SOA in Beijing before, during, and after the 2014 APEC, Atmos. Chem. Phys., 17, 981-992, 10.5194/acp-17-981-2017, 2017. Xue, J., Lau, A. K. H., and Yu, J. Z.: A study of acidity on PM2.5 in Hong Kong using online ionic chemical composition measurements, Atmospheric Environment, 45, 7081-7088, http://dx.doi.org/10.1016/j.atmosenv.2011.09.040, 2011.

Zhou, M., Zhang, Y., Han, Y., Wu, J., Du, X., Xu, H., Feng, Y., and Han, S.: Spatial and temporal characteristics of PM2.5 acidity during autumn in marine and coastal area of Bohai Sea, China, based on two-site contrast, Atmospheric Research, 202, 196-204, https://doi.org/10.1016/j.atmosres.2017.11.014, 2018.

---

## Author Response (AR1)

Dear ACP Editor,

We would like to thank the editor and the reviewers for the constructive suggestions, which are very helpful for improving our manuscript quality. We have carefully read the comments and revised the manuscript. Our responses to the two referees' comments are itemized below. All of the revisions and changes we made in the revised version are highlighted in red color.

Anything about our manuscript, please feel free to contact us via wanggh@ieecas.cn or ghwang@geo.ecnu.edu.cn .

Best regards,

Gehui Wang

08/31, 2018

**Anonymous Referee #3**

**General comments:**

**1. Comments:**

This manuscript reported the concentrations and isotopic compositions of dicarboxylic acids in the summertime of Mt. Tai. The bulk of the results presented focuses on the following aspects: (1) comparing the concentrations of species and the ratios of different dicarboxylic acids between day and night; (2) correlating the species concentrations with other parameters (T, $O_3$, RH, pH, etc); (3) the $\delta^{13}C$ of the dicarboxylic acids. The authors concluded that the dicarboxylic acids in Mt. Tai, a forested area with high elevation, mainly originate from local photochemical oxidation of biogenic emissions. Given the surrounding in Mt. Tai and sampling time, this conclusion is not surprising. Overall, the data analysis is routine and the conclusion is generally supported by experimental evidence. I recommend publication after major revisions.

**Response:** We thank the reviewer for the helpful comments above and below.

**Major Comments:**

**2. Comments:** My major concern is the lack of innovation of this study. As noted in the abstract, a few studies have performed similar measurements and analysis in the same site. The novelty of this manuscript as claimed by the authors is that the measurements were performed in summertime for the first time. It is not well justified that why it is important to conduct the measurements in summer. Also, considering the surroundings of measurement site and sampling time (i.e., summer), isn't the conclusion largely expected? It is not clear what are the scientific questions that authors aimed to address.

Response: (1) The innovation of this study is described as follows:

Firstly, several field measurements had been conducted to investigate the molecular compositions, sources and formation mechanisms of SOA including dicarboxylic acids at Mt. Tai, but most of them were performed in May and June and mainly focused on the impact of anthropogenic activities such as field burning of wheat straw, while no information on dicarboxylic acids and related SOA in Mt. Tai during the typical summertime season (i.e., July and August) is available when the emission of biogenic volatile organic compounds (BVOCs) is dominant. A 3-D model simulation shows that about 79% of oxalic acid in the global atmosphere is originated from the oxidation of natural vegetation emissions (Myriokefalitakis et al., 2011), suggesting the dominant contribution of BVOCs to the global SOA loading. Therefore, it is necessary to investigate the abundances, compositions and formation mechanisms of oxalic acid and related SOA when vegetation emission is dominant, especially in the forested highland region where aerosols are more accessible to clouds due to higher elevation (See Page 4, Line 91-105).

Secondly, compound-specific stable carbon isotope analysis is a powerful tool to provide important information of the sources and atmospheric processing of organic aerosols. Analyses of stable carbon isotope ratios of dicarboxylic acids and related SOA can be effectively applied to assessing the photochemical aging level and relative contributions of primary emissions to aerosols in the atmosphere. To our best knowledge, characteristics of the stable carbon isotopic compositions of dicarboxylic acids and related SOA in mountainous regions have not been investigated before. The current work is for the first time to report the stable carbon isotopic compositions of dicarboxylic acids and related compounds in a mountainous area, which are very helpful for improving our understanding on the sources, formation mechanisms and atmospheric behavior of SOA (Please see Page 4-5, Line 106-118).

(2) The scientific questions that we aimed to address were listed as follows:

The scientific questions that we aimed to address are sources and formation mechanisms of oxalic acid and related SOA in the free troposphere over Mt. Tai (Please see Page 5, Line 118-123).

**3. Comments:** Another major comment is regarding the comparison between daytime and nighttime. Due to the high elevation, the sampling site is in the residual layer at night. At night, the residual layer is isolated from surface emissions. The dicarboxylic acids and other species at night are mainly carried over from late afternoon formation. Also, the late afternoon mixing ratios of BVOCs, NOx and $O_3$ govern overnight chemistry within the residual layer. Please refer to Edwards et al. (2017). Assuming no further oxidation at night, the species concentrations are expected to be similar to that in the late afternoon. Thus, the day (8am-8pm) vs night (8pm-night day 8am) comparison may merely reflect the difference in species concentration between noon and 6pm (when sampling site is above the boundary layer). This study is an excellent opportunity to study the residual layer chemistry. However, in current most of discussions (Line 184, Line 223, Line 298, etc), the format is like "something is higher in daytime than nighttime, suggesting stronger photochemistry or stronger biogenic activity in the day". The authors should clearly state the physical model behind the day vs. night comparison and better explain the observations (including the $NH_4NO_3$ concentration in Line 184). The authors are encouraged to look in depth into the day vs. night comparison.

**Response:** We agree the comments above that the current work is an excellent opportunity to study the residual layer chemistry. Oxalic acid in the mountain atmosphere showed a strong linear correlation with temperature (Figure 6b) during the day and night, suggesting a dependent relationship of oxalic acid production with temperature. As the temperature was constantly variable and $O_3$ level (22±12 ppb, Table 1) was high throughout the night at Mt. Tai, thus we assume that both gas-phase and aerosol-phase oxidations should continuously proceed throughout the whole night.

As suggested by the reviewer, we did more analysis to discuss the potential factors causing the day and night differences in sulfate, nitrate and ammonium (see Figure S1 and Page 8-9, Line 223-233). Sulfate is mainly produced from aqueous phase oxidation of $SO_2$, which is favored by higher temperature and humid conditions. Thus, concentration and relative abundance of sulfate are higher in daytime than in nighttime (Table 1 and Fig. S1). Particulate nitrate is mainly formed via gas phase oxidation of $NO_2$ with OH radical and subsequent partitioning into aerosol phase with ammonia to form $NH_4NO_3$. $NH_4NO_3$ is volatile and thus lower temperature at night is favorable for $NH_4NO_3$ enriching in aerosol phase, resulting in $NH_4NO_3$ more abundant at night during the Mt. Tai observation period. These discussions were added into the revised manuscript.

**4. Comments:** Line 193-194. Why is LWC higher in daytime than nighttime?

**Response:** Aerosol LWC is controlled by the ambient relative humidity (Clegg et al., 1998) and the concentrations of inorganic salt (Fountoukis and Nenes, 2007). As shown in Table 1, the total concentration (21.7±11.5 µg m$^{-3}$) of $SO_4^{2-}$, $NO_3^-$ and $NH_4^+$ during the daytime was almost equivalent to that (20.4±8.2 µg m$^{-3}$) during the nighttime, but the relative humidity (92±5.0%) in daytime was higher than that in nighttime (77±8.2%). Therefore, the concentration of LWC in daytime was higher than that in nighttime. We have added these discussions into the revised manuscript. Please see Page 9, Line 239-245.

**5. Comments:** Line 243-248. The discussions on the higher concentration of dicarbonyls at night is confusing. Firstly, dicarbonyls are produced from photooxidation of isoprene and other VOCs, not aqueous phase reaction. Secondly, how does the "opposite pattern" suggest "aerosol aqueous phase oxidation"? Thirdly, I believe "impressed" is a typo. In light of my previous comments, the authors should explore more reasons in terms of this day vs night comparison.

**Response:** Suggestion taken. Dicarbonyls in the aerosol phase are produced by the gas-phase photooxidation of isoprene and other VOCs and the subsequent partitioning into the aerosol aqueous phase. The higher concentrations of dicarbonyls at night can in part be attributed to the nighttime lower temperatures, which are favorable for the partitioning of gaseous glyoxal and methylglyoxal into the aerosol phase. We are sorry that the "impressed" word here is a typo. We have revised the statements, please see Page 11-12, Line 302-310.

**6. Comments:** Line 268-269. Please provide evidence that $C_2/C_4$ and $C_3/C_4$ can be regarded as indicators of photochemical aging. Also, the measured $C_2/C_3$ and $C_2/C_4$ ratios are actually very close to that from vehicle exhausts. This is contradictory to the conclusion that dicarboxylic acids are mainly from biogenic emissions.

**Response:** We are sorry for the mistake about the mass ratios of $C_2/C_4$ and $C_3/C_4$ from vehicle exhausts. Kawamura and Kaplan (1987) have reported that the ratios of $C_2/C_4$ and $C_3/C_4$ from vehicle exhausts were 4.1 and 0.35, respectively (see Page 12, Line 334- 335). The $C_2/C_4$ and $C_3/C_4$ ratios in this study are $8.0 \pm 2.7$ and $1.6 \pm 0.6$, respectively, higher than those in aerosols freshly emitted from vehicle exhausts. In addition, elemental carbon (EC) concentrations were very low and similar between day and night during the observation period (Table 1), suggesting that the impact of pollutants derived from anthropogenic sources including vehicle exhausts during the campaign was negligible. Therefore, traffic emission is not expected to have a large contribution to dicarboxylic acids in this study. It's consistent with the conclusion that dicarboxylic acids in Mt. Tai during summer are mainly from biogenic emissions rather than anthropogenic emissions.

**7. Comments:** Line 279-282. This is one of many cases that the authors need to better explain the link between evidence and conclusion. Why would the "correlation between $C_2/C_4$ and temperature at night" suggest "local photochemical oxidation"?

**Response:** Decomposition of $C_4$ and/or $C_3$ into $C_2$ is one of the major formation pathways of oxalic acid, which is favored by temperature. Thus, a linear correlation of $C_2/C_4$ or $C_2/C_3$

with temperature has been frequently observed (Kawamura and Ikushima, 1993; Meng et al., 2013; Pavuluri et al., 2010). Temperature measured at the sampling site is a meteorological parameter, which only reflects the local meteorological conditions rather than the upwind conditions. Therefore, a significant correlation between $C_2/C_4$ and temperature can only be observed when SOA is largely derived from local precursor oxidation rather than from long-range transport. We have added related discussions into the text, please see Page 13, Line 341-350.

**8. Comments:** Line 360-362. The role of nighttime chemistry is not justified at all. The lack of correlation between BSOA tracers and $O_3$ at night could be attributed to that most of BSOA tracers at night are carried over from later afternoon product and the $O_3$ at night may be quickly titrated by residual NO. Again, this comment falls in the scope of day vs. night comparison.

**Response:** Thanks for your suggestion. We have revised the related contents in the revised manuscript. Please see Page 16, Line 432-441.

**Minor Comments:**

**9. Comments:** Line 61. "water" is uncountable noun. Replace "waters" with "water".

**Response:** Suggestion taken. Please see Page 3, Line 63.

**10. Comments:** Line 132. What type is the GC column?

**Response:** The type of the GC column is fused silica capillary column (HP-5, 0.2 mm × 25 m, film thickness 0.5 μm). We have added the details about the type of the GC column into the revised manuscript. Please see Lage 6, Line 153-156.

**11. Comments:** Line 304. The change in topic is too quick. Rephrase the sentence to improve the transition.

**Response:** Suggestion taken. We have rephrased the sentence as follows:

However, the average values of $C_9/C_6$ (14±9.0) and $C_9/Ph$ (7.2±2.2) at the mountaintop of Mt. Tai are higher than those in urban regions such as Xi'an, China ($C_9/C_6$: 3.1; $C_9/Ph$: 5.6) (Cheng et al., 2013) and also higher than those in other mountainous during summer such as Mt. Himalayan, India ($C_9/C_6$: 2.1; $C_9/Ph$: 0.2) (Hedge and Kawamura, 2012) and Mt. Fuji, Japan ($C_9/C_6$: 3.1) (Mochizuki et al., 2017), indicating the important contribution of biogenic sources to SOA in the Mt. Tai region. Model simulation (Fu et al., 2008) and field observations (Meng et al., 2014) have suggested that…….

Please see Page 14, Line 371-377.

**12. Comments:** Line 360. As mentioned above in the text, it is not only from O3 oxidation, also OH oxidation.

**Response:** Thanks for your suggestion. We have added "and OH radicals" in the old sentences. Please see Page 16, Line 436.

**13. Comments:** Line 377. Replace "linger" with "linear".

**Response:** Suggestion taken. Please see Page 17, Line 463.

[revised manuscript text omitted]